# Well-defined in-textile photolithography towards permeable textile electronics

Pengwei Wang[1,2], Xiaohao Ma [1,3], Zhiqiang Lin[1], Fan Chen [1], Zijian Chen[1], Hong Hu[1], Hailong Xu[1], Xinyi Zhang[3], Yuqing Shi[1,3], Qiyao Huang[1,4] ✉, Yuanjing Lin [3] ✉ & Zijian Zheng [1,2,4,5] ✉

Textile-based wearable electronics have attracted intensive research interest due to their excellent flexibility and breathability inherent in the unique three-dimensional porous structures. However, one of the challenges lies in achieving highly conductive patterns with high precision and robustness without sacrificing the wearing comfort. Herein, we developed a universal and robust in-textile photolithography strategy for precise and uniform metal patterning on porous textile architectures. The as-fabricated metal patterns realized a high precision of sub-100 μm with desirable mechanical stability, washability, and permeability. Moreover, such controllable coating permeated inside the textile scaffold contributes to the significant performance enhancement of miniaturized devices and electronics integration through both sides of the textiles. As a proof-of-concept, a fully integrated in-textiles system for multiplexed sweat sensing was demonstrated. The proposed method opens up new possibilities for constructing multifunctional textile-based flexible electronics with reliable performance and wearing comfort.

Wearable electronics hold great promise for non-invasive monitoring of physiological signals and chronic biomarkers for fitness monitoring and daily healthcare[1–8]. In the past two decades, researchers have witnessed the rapid development of flexible electronics based on a variety of thin film substrates via conventional nanofabrication techniques[9–13]. However, flexible thin-film electronics normally have relatively poor air and moisture permeability, limiting the wearing comfort, especially for long-term body status monitoring. Furthermore, the smooth surface of thin-film materials shows inadequate active sites for some applications, e.g., sensors and supercapacitors, where surface area plays an important role in the device performance[14]. Compared with thin films, textiles offer tunable three-dimensional (3D) porous structures that can provide high air and moisture permeability, large surface area, and robust mechanical softness to withstand bending, twisting, shearing, and stretching[15–20].

The integration of electronic components, such as sensors, energy modules, and data processing and transmission units, into textiles enables the creation of wearable textile electronics. These multifunctional wearables possess the capabilities to interact with the human bodies and the surrounding environments imperceptibly and comfortably, which could especially benefit real-time healthcare wearables[21–25].

The embodiment of electronics into textiles primarily requires the development of high-precision and highly conductive patterns in textiles with retained permeability, softness, and mechanical robustness, which can function as fundamental building blocks for electrical interconnects, electrodes and devices in a typical electronic system[26–32]. To date, conductive patterns are either inserted into the textiles by traditional textile technologies (e.g., weaving, knitting, or embroidery) or patterned onto the fabric surfaces via various printing

[1]School of Fashion and Textiles, The Hong Kong Polytechnic University, Hong Kong SAR, China. [2]Department of Applied Biology and Chemical Technology, The Hong Kong Polytechnic University, Hong Kong SAR, China. [3]School of Microelectronics, Southern University of Science and Technology, Shenzhen 518055, China. [4]Research Institute for Intelligent Wearable Systems, The Hong Kong Polytechnic University, Hong Kong SAR, China. [5]Research Institute for Smart Energy, The Hong Kong Polytechnic University, Hong Kong SAR, China. ✉e-mail: qihuang@polyu.edu.hk; linyj2020@sustech.edu.cn; zijian.zheng@polyu.edu.hk

techniques (e.g., inkjet printing, screen printing, and stencil printing). In the traditional textile approaches, conductive threads (e.g., silver-coated nylon yarns, stainless steel yarns, carbon nanotube yarn) are integrated into fabrics to achieve conductive patterns with typical linewidth ranging from 0.1~1 mm in textiles[33–37]. Though such an in-textile patterning approach can preserve the fabric flexibility and permeability, their fabrication processes are difficult to achieve complicated circuits in textiles due to their less compatibility with the standard electronic manufacturing processes[38]. In the printing approach, conductive inks or slurries that are prepared with a mixture of conductive materials and binders adhere to the surface of fabrics to form conductive patterns, which have been proven to be facile and versatile[39,40]. Nevertheless, due to the rough fabric surfaces and ink diffusion along the fibers within the 3D textile structures, such an on-textile patterning method is challenging to achieve a highly conductive and robust conductive patterns with precise linewidth thinner than 0.5 mm. Moreover, the conductive pastes largely covering the fabric surfaces will not only block the air and moisture permeability of textiles but also stiffen the fabrics by binding the soft fiber bundles, which may lead to cracks or even delamination with mechanical inteferences[41–44]. To increase the patterning resolution, some researchers have developed methods to first coat the rough textile surfaces with additional epoxy-like planarization layer, followed by patterning of metal electrodes. Although this approach can effectively improve the resolution, the planarization process turned the porous textile structure into a dense thin-film like substrate, and then lose the permeability and wearing comfort of textiles. In short, current state-of-the-art patterning methods for textiles show limitations in simultaneously offering conductive patterns with high precision, excellent electrical conductivity, and mechanical robustness with retained permeability and softness.

To address these challenges, we herein propose an in-textile patterning technology by combining the polymer-assisted metal deposition (PAMD) and double-sided photolithography, named in-textile photolithography, to create unique 3D interconnected well-defined and robust metal patterns in textiles. The in-textile photolithography allows for highly precise deposition of metal patterns on fabrics without the need for a binder and maintains the 3D porous structure of the textiles. Thus, the as-prepared E-textiles could maintained their desirable air and moisture permeability, flexibility and wearing comfort. The high precision for metal patterning on the fabric yarns was achieved, which overcomes the key limitations of textile pore sizes and the diffusion issue due to capillary effect in the yarn structures. Attributed to the high precision of this in-textile patterning method, fine interconnects and electrode patterns in textiles can be realized, fulfilling the requirements of commercial chip integration and contributing to high-performance and miniaturized device development. Notably, the metal patterns permeate inside the textile scaffold and exhibit electrical conduction throughout the fabric thickness within the patterned area, enabling the construction of electronics on both sides of the fabric and showing promise for multi-layer circuit construction on a single piece of textile. The as-fabricated in-textile metal patterns also deliver outstanding bending stability of 10,000 times, and washing stability of 20 times with negligible conductivity variation. As a proof-of-concept, we demonstrate the use of in-textile photolithography to fabricate an integrated multiplexed biosensing headband for wireless sweat sensing. Owing to the excellent mechanical stability and permeability of the conductive patterns in textiles, the headband realizes real-time sweat collection and the simultaneous monitoring of multiple sweat biomarkers with desirable wearing comfort. The proposed innovative approach for well-defined metal patterns with sub-100 μm resolution in textiles will pave the way towards permeable, high-performance, multifunctional, and highly integrated textile electronics for practical wearable applications such as non-invasive monitoring and human-machine interfaces.

## Results

### In-textile photolithography of conductive metal patterns in textiles

The fabrication of well-defined conductive metal patterns in textiles via utilizing in-textile photolithography is illustrated in Fig. 1a, which consists of two steps: 1) deposition of metal in textiles via PAMD and 2) formation of metal patterns in textiles via double-sided photolithography technique. Briefly, the pristine fabric was first metalized by the solution-processable PAMD approach, through which fibers in the textile scaffold were coated with a thin layer of metal. In the PAMD approach, the fabric was grafted with a thin layer of interfacial polymer containing ligands for the immobilization of catalytic ions, followed by the electroless deposition of a metal layer on the modified fiber surfaces. It is worth mentioning that a metal/polymer composite structure was formed at the interface between the fiber surface and the metal layer, which has been proven to facilitate the adhesion stability of the as-deposited metal layers on the textile substrates (Supplementary Fig. 1)[45–47]. After PAMD, the metalized fabric was dip-coated with the negative photoresist, followed by ultraviolet (UV) exposure. Unlike the traditional photolithography applied in thin-film microelectronic fabrications, which typically involved one-sided UV exposure, both sides of the photoresist-coated metalized fabric were covered with identical photomasks and simultaneously exposed to UV light sources (Inset in Fig. 1a). Such a double-sided UV exposure could ensure that all the photoresist layers loaded on fibers in the 3D textile structure were sufficiently subjected to the UV light, enabling the transfer of the pre-designed pattern from the photomask to the photoresist layer in the fabric. After double-sided UV exposure, the non-exposed photoresist coating was selectively removed in the development process and the metal layer beneath was then etched away. Meanwhile, the reacted photoresist coating that had been exposed to UV light could prevent the metal layer beneath from being removed during the etching step (Supplementary Fig. 2a, b). Upon the removal of the reacted photoresist coating, the designed metal patterns were precisely formed in the fabric (Supplementary Fig. 2c, d).

By utilizing the in-textile photolithography technology, one could precisely fabricate arbitrary conductive patterns in the fabric substrate on a large scale. Figure 1b demonstrates the copper (Cu)-based printed circuit board (PCB) with a dimension of 7 cm × 10 cm in a piece of polyester fabric. Cu tracks with distinct track boundaries, ranging in linewidths from 300 μm to 2 mm, were permeated into the textile scaffold. The scanning electron microscopy (SEM) images reveal that the Cu only conformally wrapped around the fiber surfaces within the designated patterned regions without binding the fiber bundles, which could preserve the 3D fibrous structure of the textile (Fig. 1c, d). As a result, the patterned fabric exhibited similar softness and water vapor permeability as the pristine fabric (Fig. 1e). Thanks to the versatility of PAMD, in-textile photolithography offered the potential to achieve diverse conductive patterns made with different metals in a wide range of textile substrates. As a proof-of-concept, we fabricated the silver (Ag) circuit interconnects (Fig. 1f) and nickel (Ni) interdigitated electrodes in the polyester fabrics (Fig. 1g), Cu sensing electrode array in the glass-fiber and cotton fabric (Fig. 1h, Supplementary Fig. 3 and Fig. 4a), and Ni electrodes in the polypropylene (PP) non-woven fabric (Fig. 1i and Supplementary Fig. 4b). The 100 μm patterning resolution of the Ag circuit interconnects and Ni interdigitated electrodes in the polyester fabrics was achieved, demonstrating the wide applicability of in-textile photolithography as a universal method in the creation of well-defined, multifunctional, and highly integrated circuits for textile electronic applications.

### Permeable, robust, and well-defined conductive metal patterns in textiles

In-textile photolithography outperformed the conventional on-textile patterning methods in terms of fabric permeability, robustness,

patterning resolution, and electrical resistance. Such exceptional performance could be attributed to the permeated conductive patterns within the textile scaffold and their conformal and firm metal coating that only surrounded the individual fibers (Fig. 2a and Supplementary Fig. 5a), which were enabled by the solution-processed metal deposition technique (PAMD). In this method, fiber bundles in the patterned region could retain their original fibrous configuration, thereby preserving the 3D porous structure that was favorable for air and moisture permeation as well as the textile softness. More importantly, the composite structure formed at the metal/fiber interface, as illustrated in the above session, played a crucial role in establishing interfacial bonding and enhancing the mechanical durability of the conductive patterns on the fabric substrate[46,47]. On the contrary, due to the diffusion and penetration of the printing ink into the fabric scaffold, conductive coating in the conventional on-textile patterning approach (screen printing was demonstrated in this work) mostly covered the surface of the textile and bound the fiber bundles in the printed region (Fig. 2b and Supplementary Fig. 5b). This would severely block the pores in the textile scaffold and stiffen the fabric,

which might consequently affect not only the transmission of air and moisture but also the mechanical flexibility of the conductive patterns. As a result, the fabric patterned by in-textile photolithography exhibited a 21-fold higher air permeability (77.3 mm/s) compared to the one fabricated through the on-textile patterning method (3.4 mm/s) and possessed a 10% enhancement in water vapor permeability (561 g/m$^2$/day) (Fig. 2c). With the same patterning resolution of 500 μm, the in-textile photolithography approach provided superior robustness to the conductive tracks in the fabric upon mechanical deformation, compared to the on-textile patterning approach (Fig. 2d). The resistance change ($R/R_0$; R: resistance after the bending test; $R_0$: resistance before the bending test) of the metal pattern made by in-textile photolithography increased to ~2 at the first 500 bending cycles (bending radius: 4.7 mm) and remained relatively constant thereafter. However, the pattern made by the on-textile patterning method experienced a significant resistance change of more than 4 after 10,000 bending cycles.

Most importantly, in-textile photolithography inherited the high precision of the conventional photolithography technique, enabling

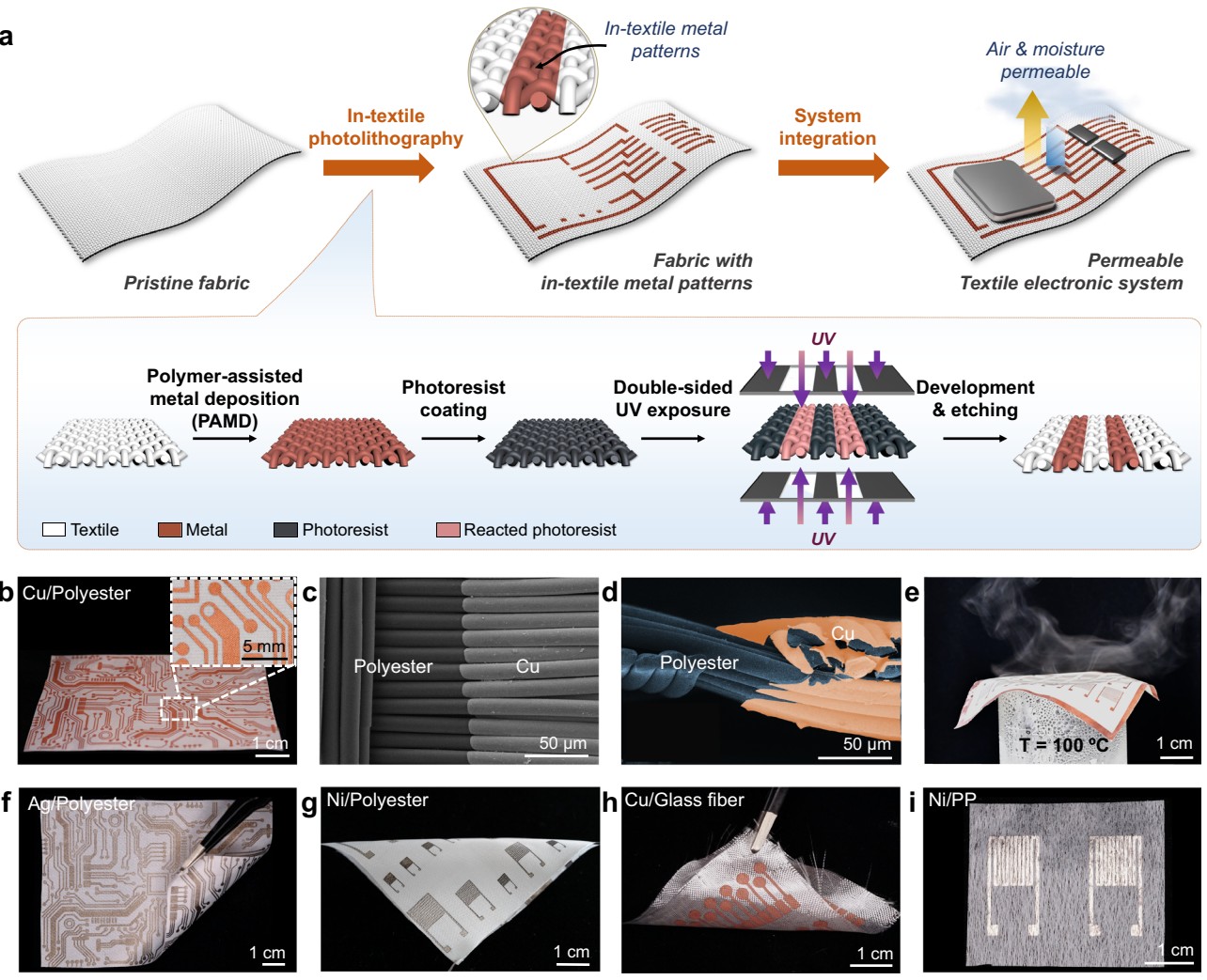

**Fig. 1 | In-textile photolithography for conductive metal patterns in fabrics.** **a** Schematic illustration showing the development of textile electronics by in-textile photolithography technology. **b** Digital image of the Cu PCB circuit patterned in polyester fabric. Inset is the high-resolution image showing the Cu tracks in the fabric. **c** SEM image showing the boundary of the Cu pattern in polyester fabric. **d** Cross-sectional SEM image showing the boundary of the Cu pattern in polyester fabric. The numbers of experimental repetitions of **c** and **d** were five times,

respectively. **e** Digital image showing the water vapor permeability and softness of the fabric patterned with Cu patterns. **f** Digital image of the Ag PCB circuit patterned in polyester fabric. **g** Digital image of the Ni interdigital electrodes with different scales patterned in polyester fabric. **h** Digital image of the Cu sensing electrode arrays patterned in glass-fiber fabric. **i** Digital image of the Ni electrode patterns in non-woven PP fabric.

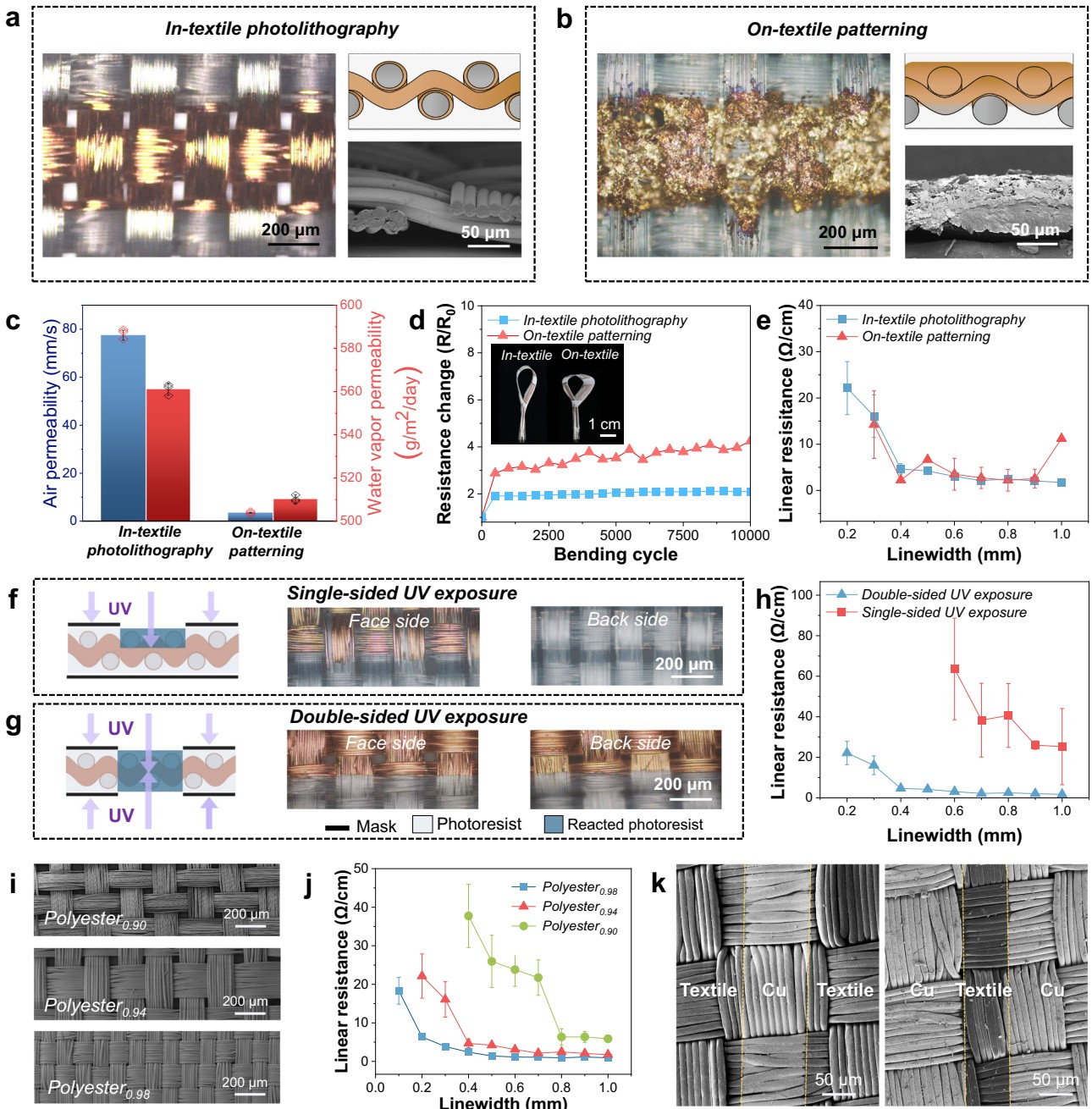

**Fig. 2 | Characterization of well-defined and conductive metal patterns in textiles. a** Optical image (left), cross-sectional schematic diagram (top right), and SEM image (bottom right) of the Cu pattern fabricated by the in-textile photo-lithography method. **b** Optical image (left), cross-sectional schematic diagram (top right), and SEM image (bottom right) of the Cu pattern fabricated by on-textile patterning method (screen printing). **c** Comparison of water vapor and air per-meability of the fabrics coated with Cu made by the in-textile photolithography method and the on-textile patterning method. Error bars represent the s.d. of the mean from three fabrics coated with Cu. **d** Comparison of resistance changes (R/$R_0$) of the Cu patterns made by the in-textile photolithography method and on-textile patterning method. Inset is the digital image showing the flexibility of the Cu patterns made by two methods. **e** Linear resistance of the Cu patterns as a function of the patterning resolution. Error bars represent the s.d. of the mean from three Cu patterns. **f** Schematic diagram of the single-sided UV exposure (left) and optical images of the face side and back side of the Cu pattern made by in-textile photo-lithography (right). **g** Schematic diagram of the double-sided UV exposure (left) and optical images of the face side and back side of the Cu pattern made by in-textile photolithography (left). **h** Linear resistance of the Cu patterns as a function of the patterning resolution. Error bars represent the s.d. of the mean from three Cu patterns. **i** SEM images of the Polyester$_{0.90}$, Polyester$_{0.94}$, and Polyester$_{0.98}$ fabrics. 0.90, 0.94, and 0.98 are the fabric cover factors of the corresponding fabrics. **j** Linear resistance of the Cu patterns as a function of patterning resolution. Error bars represent the s.d. of the mean from three Cu patterns. **k** SEM images showing the precise Cu pattern with 100 μm linewidth (left) and 80 μm interspace between Cu patterns (right) in Polyester$_{0.98}$ fabrics.

the creation of precise conductive patterns with sharp and well-defined boundaries. While the conventional on-textile patterning approach could only achieve a conductive track with a minimal line-width of 300 μm, in-textile photolithography surpassed this limitation by ensuring the promising electrical conductivity (22 Ω/cm) even for the conductive track with a linewidth of as fine as 200 μm (Fig. 2e). Such significant enhancement of patterning resolution was attributed not only to the protection of the photoresist coating that could

effectively mitigate the diffusion of conductive coating but also to the implementation of the double-sided UV exposure mechanism in the in-textile photolithography process. To study the crucial role of double-side UV exposure in attaining well-defined conductive metal patterns in the textile scaffold, we compared the coating morphologies and electrical resistances of the metal patterns that were respectively subjected to single-sided and double-sided UV exposures during the fabrication. Due to the non-transparency of the textile materials, when only one side of the photoresist-coated metallic fabric was covered with a photomask of the desired pattern and subjected to UV exposure, the UV light was unable to penetrate the opposite side of the fabric. As a result, the photoresist coating on the opposite side of the fabric could not undergo a complete crosslinking reaction, leading to its dissolution in the development step and the exposure of its underneath metal layer. Since the exposed metal layer was etched away in the subsequent etching step, the conductive track became discontinued throughout the textile scaffold (Fig. 2f and Supplementary Fig. 5c). In contrast, double-sided UV exposure could ensure that the photoresist coated around the fibers in the fabric scaffold was sufficiently exposed to the UV light and underwent a complete chemical reaction, thereby providing a strong protective barrier in the subsequent development and etching steps. Consequently, the metal track beneath the reacted photoresist within the designated pattern area was preserved (Fig. 2g and Supplementary Fig. 5d). Such preservation endowed the conductive pattern with high integrity, exceptional patterning resolution and precision, as well as excellent electrical conductivity. The thickness of our as-prepared textile is 100 μm and the UV exposure depth is approximately 80 μm, a thickness that ensures the integrality of the Cu pattern after double-sided photolithography. Thicker textiles like Polyester with a thickness of 170 μm created through the single-sided UV exposure exhibited an average thickness of 80 μm (Supplementary Fig. 6). Noted that the textile within a thickness of 160 μm is suitable during the double-sided photolithography including our as-prepared textile. As depicted in Fig. 2h, conductive patterns created through the single-sided UV exposure exhibited the linear resistance of 25~63 Ω/cm when their linewidths fall within the range of 600 μm to 1000 μm. These patterns lost their electrical conduction when their linewidths were finer than 600 μm. Conversely, patterns created by double-sided UV exposure in the photolithography step demonstrated remarkably low linear resistance of 1.7-4.6 Ω/cm when the linewidths were designed within the range of 400 μm to 1000 μm. The patterning resolution can be further refined to 200 μm while still maintaining exceptional electrical conductivity (~22 Ω/cm).

Furthermore, the extent of yarn coverage in the woven fabric, which could be expressed by the fabric cover factor (i.e., the ratio of the fabric area covered by warp and weft yarns to the projection area of the fabric), determined the patterning resolution of the in-textile photolithography as well as the electrical performance of the conductive metal patterns. With the increase of the fabric cover factor, yarns in the fabric became closer to each other, offering a substrate with higher yarn coverage. This increased yarn coverage can ensure sufficient conductive contacts along the fiber bundles within the patterned region, offering electrical conduction to the metal pattern even with a fine linewidth. For instance, when using a woven fabric substrate with a cover factor of 0.90 for the in-textile photolithography, the achievable conductive metal pattern was limited to a minimal line-width of 400 μm (Fig. 2i, j and Supplementary Fig. 7a–c). However, on a woven fabric with a cover factor of 0.98, it became possible to reach a 100 μm-width metal pattern with a linear resistance of lower than 18 Ω/cm. More importantly, the distance between two 100 μm-width metal patterns could be as fine as 80 μm (Fig. 2k). To the best of our knowledge, this is the highest patterning resolution achievable on fabric substrate (Supplementary Table 1). It is also worth highlighting that with such a high yarn coverage, the conductive fabric still maintained a high water vapor permeability (591 g/m²/day) and air permeability (241 mm/s) (Supplementary Fig. 7d), which signified the immense potential of the in-textile photolithography technique in producing permeable electronic textiles suitable for long-term wearable and on-skin applications.

## High-performance electronics enabled by robust and well-defined conductive metal patterns in textiles

The in-textile photolithography is capable of developing metal patterns with various linewidths in fabrics, opening opportunities for constructing textile-based electrodes and devices. Since robustness is highly essential for wearable applications, the physical and electrical stability of the metal patterns was examined under various deformations. As shown in Fig. 3a, the tensile strain at break of the polyester fabric after in-textile photolithography process has decreased about 11% compared to the original commercial polyester fabric, which indicates that the physical properties of the textiles can be well-maintained. A series of Cu patterns with linewidths ranging from 400 μm to 700 μm in the polyester fabric were subjected to the 10,000-cycle bending test at the bending radius of 4.4 mm. As shown in Fig. 3b, the as-made Cu patterns only showed a slight resistance increase in the first 1000-2000 bending cycles and then remained stable afterward. The observed increase in resistance in the initial stage of the bending test could be attributed to the displacement of the metalized fibers within the patterned region, which was caused by mechanical deformations. Such a displacement tended to reach an equilibrium state with repeated bending, leading to a stabilized resistance level afterward. The robustness of the patterned conductive fabric could also be reflected by its electrical stability under the washing condition. Despite the high-speed agitation during the washing, which could introduce significant mechanical instability to the washed subjects, the electrical conductivity of the washed Cu patterns remained stable. After 20 cycles of washing, the resistance of the Cu tracks patterned in the fabric increased by only ~2 folds, exhibiting the excellent durability of the conductive metal patterns made by in-textile photolithography (Fig. 3c). Notably, the robustness of these conductive Cu patterns could be reinforced by employing additional electrodeposition of stable metals such as gold (Au). The presence of this protective barrier served to enhance the pattern's ability to endure stress and strain during the mechanical deformations: the resistance of the Au-deposited Cu pattern (Au@Cu) in the fabric did not show obvious change during the 10,000-cycle bending test (Fig. 3b), and only exhibited a marginal resistance increase up to 20 washing cycles (Fig. 3c). To further validate the robustness of the in-textile integrated circuits, washability and mechanical stability were characterized. With Ecoflex encapsulation, the resistance of the interconnects measured by connecting to 0 Ω resistors did not show obvious change (11% variation) within the 120 cycles of crumping test (Fig. 3d). It is also observed that the interconnect resistance variation can be reduced to 50% after 20 cycles of washing (Fig. 3e).

Since the metal patterns made by in-textile photolithography permeated inside the textile scaffold and exhibited electrical conduction throughout the patterned area, they could function as the fundamental building blocks for the construction of electronic devices. Specifically, these well-defined conductive patterns could benefit the development of high-performance miniatured electronic devices. We developed interdigital electrodes for micro-supercapacitors as a concept demonstration. Compared to the on-textile patterning approach (such as screen printing) with limited resolution of 400 μm and resulted thin film materials coverage on one side of the electrodes on the fabric, the in-textile photolithography technique could create interdigital electrodes that preserve the unique three-dimensional porous structures. Uniform and precise metal coating with both line-width and their interspace down to 200 μm could ensure successful device fabrication without short circuit issue among electrodes. Based

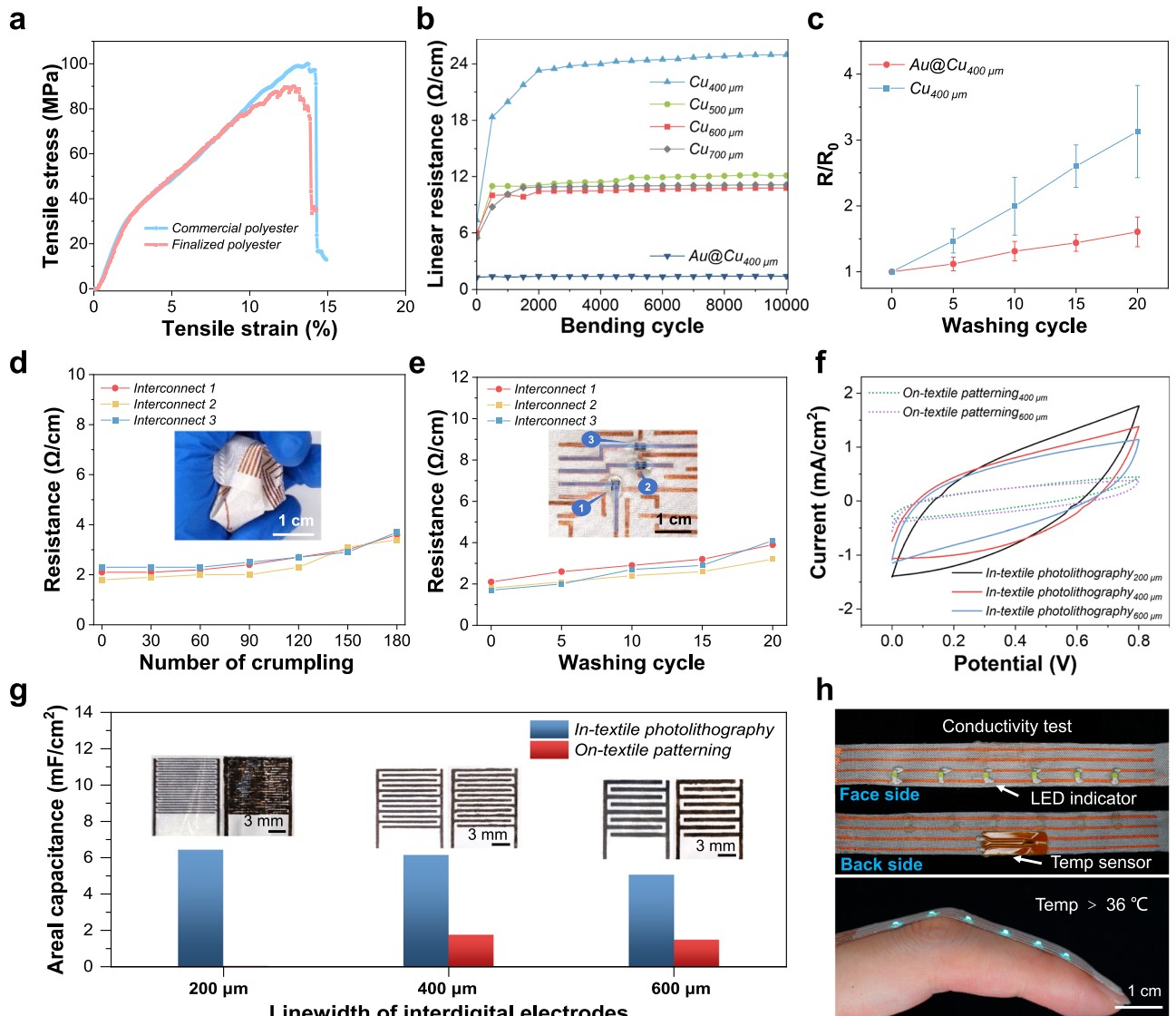

**Fig. 3 | High-performance electronics enabled by robust and well-defined conductive metal patterns. a** Tensile property of the commercial polyester fabric and polyester fabric after in-textile photolithography. **b** Resistance of the Cu patterns with different linewidths in polyester fabric during the 10,000-cycle bending test (bending radius: 4.4 mm). **c** Resistance changes of the Cu patterns with and without additional Au deposition upon 20 washing cycles. Error bars represent the s.d. of the mean from five Cu patterns. **d** Resistance of the interconnects during 180 times crumpling test. **e** Resistance upon 20 washing cycles (encapsulated with Ecoflex and measured by connecting the interconnects to 0 Ω resistors). **f** Cyclic voltammetry curves of the micro-supercapacitors made with interdigital Ni electrodes with different linewidths. $MnO_2$ is the electrochemically active material. **g** Areal capacitance of the micro-supercapacitors with different linewidths. Insets are digital images showing the electrode arrays of the micro-supercapacitors with different linewidths. **h** Digital images of the double-sided wearable temperature monitoring patch with in-situ alarming function based on well-defined and double-sided Cu pattern in polyester fabric.

on theoretical calculation, with such conformal metal coating on the textiles, the surface area is estimated to be 370% larger than the thin films formed via screen printing. As a result, under the same footprint area, the in-textile interdigital electrodes could offer a larger surface area for the conformal deposition of electrochemically active materials (i.e., manganese dioxide ($MnO_2$) in this work). The as-fabricated manganese dioxide ($MnO_2$) micro-supercapacitors deliver significantly areal capacitance enhancement of more than 300% compared to those made by screen printing under the same electroplating conditions (Fig. 3f, g and Supplementary Fig. 8). Furthermore, benefiting from the electrical conductivity provided by the metal patterns throughout the fabric thickness, electronic devices could effectively utilize both sides of the fabrics for multi-layered circuit construction on a single piece of textile. As a proof-of-concept, a wearable temperature monitoring patch with an in-situ alarming function was designed (Fig. 3h and

Supplementary Movie 1). The sensor was embedded on the back side of the fabric that was next to the skin for reliable epidermal temperature sensing, while the light-emitting diodes (LEDs) on the face side of the fabric were made for alarm when the measured temperature exceeded 36 °C. Such a double-sided conductive pattern effectively reduced the complexity of multi-layered electronic system design on the textile, demonstrating its prospects for practical applications.

## Fully integrated biosensing headband for multiplexed sweat monitoring

Textile-based electronics could provide superior permeability and flexibility, which is highly desirable for wearable sweat sensing[48–54]. However, the rough surface of the textile often leads to less controllability and uniformity for active materials deposition to construct

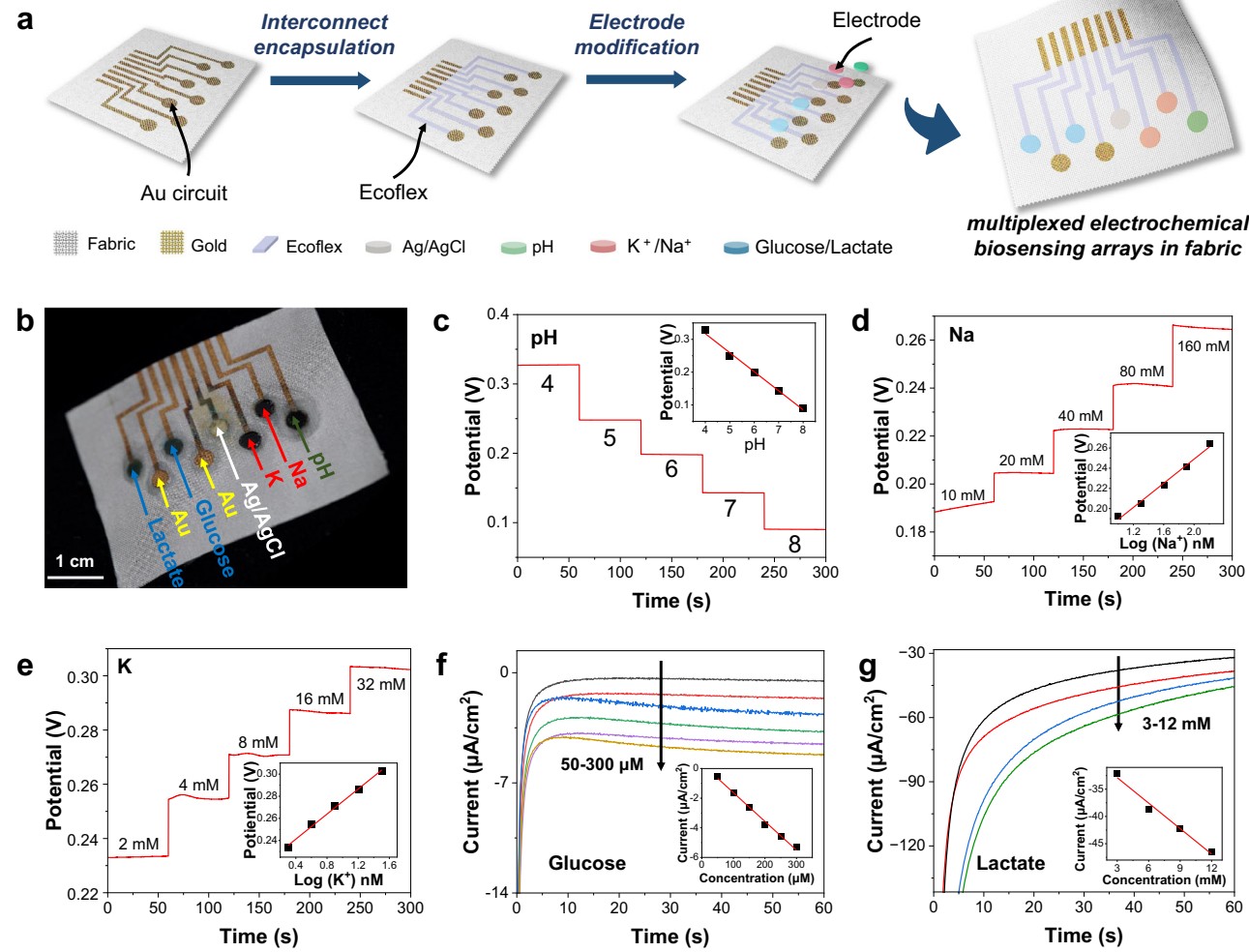

**Fig. 4 | Fabrication and performance evaluation of the multiplexed biosensor array for sweat monitoring. a** Schematic illustration showing the fabrication process of the biosensor array in fabric. **b** Digital image of as-prepared biosensor array in fabric. **c–g** Sensing performance of as-made pH, Na⁺, K⁺, glucose, and lactate sensors.

high-performance biosensors. In this regard, the continuous and uniform metal coating on the fibers achieved via the as-developed in-textile photolithography provided a desirable textile platform for biosensor fabrication. As shown in Fig. 4a, multiplexed electrochemical biosensing arrays including ion-selective potentiometric sensors for pH, sodium ($Na^+$), and potassium ($K^+$) analysis, as well as enzymatic amperometric sensors for glucose and lactate tracking, were designed and fabricated on the Cu patterned fabric. To improve the electrode durability during further modification and practical utilization, the produced Cu electrode and interconnect patterns were firstly electroplated with Au, and then the uncovered Cu areas were completely etched with ferric chloride ($FeCl_3$). Although the conductivity of the Au@Cu electrode slightly decreased after etching, it exhibited improved bending stability (Supplementary Fig. 9a–c). The interconnects were additionally encapsulated with Ecoflex to prevent short circuits in the constructed circuits during the sweat sensing. It is worth mentioning that such encapsulation preserved the porosity of the textile electronics without forming a rigid layer, thereby maintaining permeability and bending stability (Supplementary Fig. 9d–f). Afterward, the electrode areas were sequentially functionalized with corresponding active materials to obtain a highly selective multiplexed biosensor array for sweat monitoring. To minimize the sensor array, a two-electrode configuration was adopted for each type of sensor (Fig. 4b). Specifically, for the potentiometric pH sensor, the pH sensing

responses were characterized based on the dynamic open circuit potential (OCP) readouts between the polyaniline (PANI) functionalized working electrode and silver/silver chloride/polyvinyl butyral (Ag/AgCl/PVB) reference electrode (Supplementary Figs. 10a, 11, 12a–c). For the potentiometric $Na^+$ and $K^+$ ion sensors, their working electrodes were constructed by sequentially depositing poly(3,4-ethylenedioxythiophene) polystyrene sulfonate (PEDOT: PSS) and corresponding ion-selective membranes onto the Au@Cu electrode areas (Supplementary Figs. 10b and 12d–f), while their Ag/AgCl/PVB reference electrodes were shared with the pH sensor. For the amperometric enzymatic sensors for glucose and lactate analysis, their selective working electrodes were attributed to the glucose and lactate oxidase immobilized on the Prussian blue (PB) as ion-electron transducing layers (Supplementary Fig. 12g–i). The enzymatic working electrodes were then coupled with Au counter electrodes to extract current outputs (Supplementary Fig. 10c).

The performance of these electrochemical sensors based on the Au@Cu conductive patterns in the fabrics was studied. The as-prepared pH sensor produced steady signals with an average sensitivity of 54.93 mV/decade in the range of pH 4 to 8 (Fig. 4c). The sensors for $Na^+$ and $K^+$ ion analysis delivered the average sensitivities of 59.73 mV/decade in response to 10 mM to 160 mM $Na^+$, and 56.61 mV/decade in 2 mM to 16 mM $K^+$ in the artificial sweat, respectively (Fig. 4d, e). The sensitivities of these potentiometric sensors matched well with the

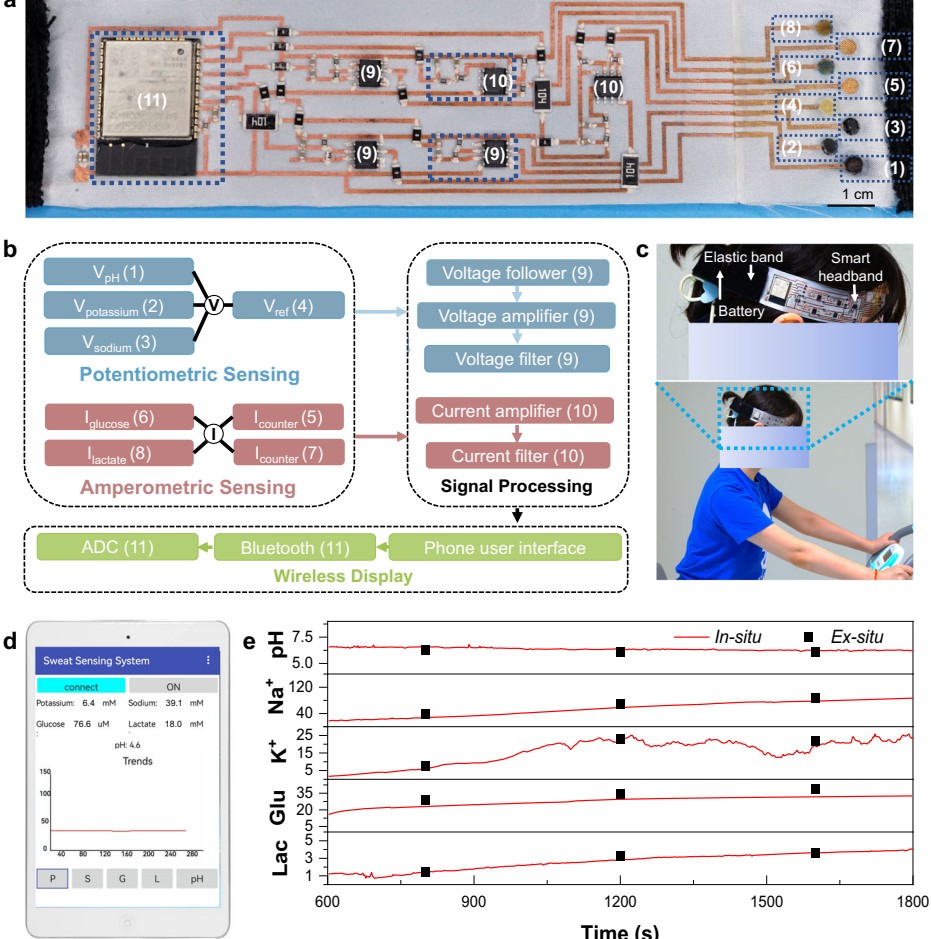

**Fig. 5 | Design and on-body application of the integrated in-textile headband.** **a** Digital image of the in-textile headband integrated with multiplexed sensing arrays and data analysis transmission circuit. **b** The logic flow of the in-textile headband. **c** Digital image showing in-situ sweat monitoring of the in-textile headband during exercise (the headband is mounted on the forehead). **d** Custom-designed mobile application for real-time and continuous biomarker sensing. **e** Representative real-time in-situ sweat biomarkers concentration monitoring during endurance cycling with ex-situ validation via analytical tools (pH meter, ICP-MS, and HPLC).

theoretical values according to Nerst equations[55,56]. The glucose sensors performed a sensitivity of 19.43 nA/μM/cm² in the ranges of 50 μM to 300 μM (Fig. 4f), while the lactate sensors showed 1.59 μA/mM/cm² in 3 mM to 12 mM (Fig. 4g). The fabrication processes exhibit high feasibility to the textile platforms, and the as-prepared sensor arrays have desirable reproducibility as shown in Supplementary Fig. 13. These remarkable factors could be mainly attributed to the large surface areas of the 3D porous textile structures and precise metal patterning in the fabric enabled by the in-textile photolithography.

The as-fabricated multiplexed sensing array was then integrated with an in-textile circuit into a headband for the real-time and wireless sweat monitoring demonstration (Fig. 5a and Supplementary Fig. 14). The circuit design was based on the one-layer configuration to maintain the permeable fabric structure while ensuring conductivity (Supplementary Fig. 15). The schematic diagram of the circuit design and the flow chart are illustrated in Fig. 5b. A high-impendence potential readout circuit for potentiometric sensors and a high-precision current amplification circuit for amperometric sensors were designed to realize the signal extraction while eliminating interference signals. The extracted analog signals were processed through an amplifier and filter before converting to digital signals with an ESP32 chip. The chip also integrated a Bluetooth module for data wireless transmission to mobiles (Supplementary Fig. 16).

Real-time physiological monitoring was performed on a volunteer wearing the integrated in-textile sensing headband during constant-load exercise on a cycle ergometer (Fig. 5c). To evaluate the wearing safety, we conducted a long-term on-skin test. As shown in Supplementary Fig. 17, the as-fabricated sensor electrodes did not show observable effects on the skin while other traditional flexible substrates, such as polyester film, led to skin erythema possibly due to the impermeable structure and large adhesion strength. After a 10-min warm-up, sweat biomarker levels were continuously monitored for the following 20 min cycling, along with sweat samples collection for ex-situ data validation. The sweat concentrations can be wirelessly tracked on the custom-designed mobile App (Fig. 5d and Supplementary Movie 2). The accuracy of on-body measurements was verified by comparing extracted signals with ex-situ measurements of collected sweat samples by pH meter, inductively coupled plasma mass spectrometry (ICP-MS), and high-performance liquid chromatography (HPLC) (Fig. 5e), which showed a consistent tendency. The estimated relative standard deviation (RSD) for the sensing results were 3.4% for pH, 19.3% for Na⁺, 8.7% for K⁺, 21.2% for glucose, and 6.3% for lactate, respectively, indicating the potential of such an integrated in-textile sensing system fabricated by the in-textile photolithography technology for non-invasive human healthcare.

## Discussion

In summary, we present an innovative approach, in-textile photo-lithography, to effectively address the challenges associated with conductive patterning in 3D porous textiles. In-textile photolithography enables a high patterning resolution of 100 μm with high electrical conductivity, outstanding bending stability of 10,000 times, and washing stability of 20 times, as well as well-preserved air and moisture permeability for conductive metal patterns in textile fabrics. Such a high-precision patterning method allows for the realization of fine electrodes with sufficient surface areas within the porous textiles, facilitating the fabrications of high-performance and miniaturized devices as well as the seamless integration with commercial chips for circuit development. As a result, functional electronic devices, such as high-performance micro-supercapacitors and temperature-sensing patches with in-situ alarm functions, can be constructed in the fabric, which significantly exhibits the promise of multi-layer circuit construction on a single piece of textile. Importantly, we develop the integrated multiplexed biosensing headband for wireless sweat sensing, as a proof-of-concept demonstration with the use of in-textile photolithography. Owing to the excellent mechanical robustness and permeability of the well-defined conductive patterns in textiles, the headband realizes real-time sweat collection and simultaneous monitoring of multiple sweat biomarkers with desirable wearing comfort. The proposed innovative approach for high-precision metal patterns in textiles sheds light on the advancements of permeable, high-performance, multifunctional, and highly integrated textile electronics for wearable applications such as non-invasive health monitoring and human-machine interfaces.

## Methods

### In-textile photolithography of conductive metal patterns in textiles

The fabrication of conductive metal patterns in textiles via in-textile photolithography consists of two steps: 1) deposition of metal in textiles via polymer-assisted metal deposition (PAMD), and 2) formation of metal patterns in textiles via double-sided photolithography technique. Before the PAMD step, the pristine polyester fabric was cleaned by using deionized (DI) water and ethanol, followed by being dried at 80 °C for 15 min. Afterward, the cleaned fabric was treated with 2 M NaOH solution at 80 °C for 2 h to render the hydrophilicity. In the PAMD step, the NaOH-treated fabric was first immersed into a mixture of 2 mL acetic acid, 8 mL DI water, 8 mL trimethoxysilane, and 190 mL ethyl alcohol at room temperature for 1 h for silanization. The silanized fabric was then immersed into an aqueous solution of [2-(methacryloyloxy)-ethyl] trimethyl ammonium chloride (METAC) (20% v/v in DI water, 100 mL) and potassium persulfate (400 mg) at 80 °C for 2 h to graft the polymer brush (PMEATC) to the fiber surfaces. Then the PMETAC-coated textiles were immersed into $5 \times 10^{-3}$ M $(NH_4)_2PdCl_4$ aqueous solution at room temperature for 30 min for the loading of catalytic moieties $[PdCl_4]^{2-}$. Finally, the $[PdCl_4]^{2-}$-loaded fabric was immersed into the electroless plating bath of targeting metal to deposit the metal layer onto the fiber surfaces. For Cu coating, the $[PdCl_4]^{2-}$-loaded fabric was submerged into a bath containing a 1:1 (v/v) mixture of solution A and B for 50 min to complete the Cu deposition. Solution A was an aqueous mixture of NaOH (12 g/L), $CuSO_4 \cdot 5H_2O$ (13 g/L), and $KNaC_4H_4O_6 \cdot 4H_2O$ (29 g/L) in DI water. Solution B was a 9.5 mL/L formaldehyde aqueous solution. For Ni coating, the fabric was immersed in the Ni bath for 50 min. The Ni bath contained the aqueous solution of 40 g/L $Ni_2SO_4 \cdot 5H_2O$, 20 g/L sodium citrate, 10 g/L lactic acid, and 1 g/L dimethylamine borane. The pH value of the solution was adjusted to 7.5 by using ammonia solution before use.

In the second step of double-sided photolithography, the metal-coated fabric was first dip-coated with photoresist (NR9-1500p, Futurrex, USA), and then was heated in an oven at 110 °C for 10 min to solidify the photoresist layer covered on the fabric. Both sides of the photoresist-coated metalized fabric were subsequently covered with two identical photomasks and subjected to double-sided UV exposure for 3 min, followed by heating in an oven at 100 °C for 3 min to promote the photoresist reaction. The photoresist in the non-patterned area was then removed by a developer solution (RD6, Futurrex, USA), and the metal beneath the non-patterned area was etched by an etching solution ($FeCl_3$). Finally, the photoresist on the metal pattern was washed away by dimethylsulfoxide (DMSO). After washing with alcohol and DI water, fabric patterned with conductive metal tracks was obtained. In comparison, the metal-coated sample was also exposed to a one-sided UV light source with only one mask covered on one side of the fabric (i.e., single-sided UV exposure). To study the differences in patterning resolution and electrical conductivity of the conductive patterns made by in-textile photolithography and conventional on-textile patterning techniques, screen printing of conductive patterns on fabric was demonstrated as well. The fabric was covered with a screen (200 mesh count) of the desired patterns and fixed with a screen printer. The Cu ink (Weixiulao, China) was then brush-coated onto the screen with a squeegee and transferred through the mesh to the fabric substrate to create the desired patterns. The printing speed was controlled as 10 mm/s and the printing angle was set around 45°.

### Fabrication of micro-supercapacitors

The electrode fabrication of micro-supercapacitors was based on in-textile photolithography and screen printing, respectively. The $MnO_2$ deposition solution included 0.05 M $Mn(CH_3COO)_2$, 0.05 M $Na_2SO_4$, and 10% ethanol. A periodic voltage wave of 10% duty cycle (an upper voltage of 1.5 V) and 90% of 0.7 V with a frequency of 0.1 Hz was applied for the deposition of $MnO_2$ onto the interdigital electrodes in the fabric for 10 mins. The electrolyte was prepared by dissolving 12 g of PVA and 12 g of 85 wt.% $CH_3COOLi$ in 120 mL DI water at a temperature of 85 °C. After that, the electrolyte was cooled down to room temperature and then pasted onto the electrode areas to form the micro-supercapacitor.

### Surface area ratio simulation and calculation of the micro-supercapacitors

One elliptical conductive weft yarn (radius of ellipse: 2.5/1) has 12 semicircular conductive fibers while one elliptical conductive warp yarn (radius of ellipse: 3/1) has 12 semicircular conductive fibers pattered in $Polyester_{0.98}$ fabric. The ratio calculation equation can be written as follows:

$$Ratio = \left( \frac{L_{warp} \times \frac{\pi}{2}}{L} + \frac{L_{warp} \times \frac{\pi}{2}}{L} \right) \times 0.98$$
$$= \left( \frac{(\pi + 2 \times 2)\frac{\pi}{2}}{2 \times 3} + \frac{(\pi + 2 \times 3)\frac{\pi}{2}}{2 \times 4} \right) \times 0.98 = 3.72 \tag{1}$$

where $L$ is the length of the cross section for one chosen yarn, $L_{warp}$ and $L_{weft}$ are the perameters of chosen yarn, 0.98 is the cover factor of the fabric.

### Fabrication of biosensor array for sweat monitoring

A Cu pattern with 8 electrodes was designed and fabricated in the polyester fabric for the construction of $Na^+$, $K^+$, pH, glucose, and lactate sensors. Before the electrode modification, the Cu pattern was coated with Au via two-electrode electrodeposition with 1 mA/cm² in Au plating solution (Caswell, Inc, USA) for 90 min. Afterward, the pattern was etched with 1 M $FeCl_3$ to remove the exposed Cu. The interconnecting section was then encapsulated by Ecoflex (00-50, Makesly).

$Na^+$, $K^+$, and pH sensors were constructed based on a two-electrode configuration, where the Au-deposited Cu (Au@Cu) electrodes were employed as building blocks for working electrodes and

Ag/AgCl reference electrodes. The Ag/AgCl reference electrode was fabricated by electrodepositing the Ag onto the Au@Cu electrode at a constant current of $1\,mA/cm^2$ for 30 min in the Ag plating solution (Caswell, Inc, USA), and followed by the etching process with 0.1 mol/L $FeCl_3$ for 60 s. 2 μL of PVB reference solution was then drop cast onto the Ag/AgCl layer to form Ag/AgCl/PVB reference electrode. The PVB reference was prepared by dissolving 79.1 mg PVB and 50 mg NaCl into 1 mL methanol. For the working electrode of the $Na^+$ and $K^+$ sensors, PEDOT: PSS was adopted as the ion-electron transducer to reduce the potential drift of the ion-selective electrode. It was first electrodeposited onto the Au@Cu working electrode by using the plating solution of 0.01 M 3,4-Ethylenedioxythiophene (EDOT) and 0.1 M poly(sodium 4-styrenesulfonate) NaPSS at a constant current of $2\,mA/cm^2$ with 20-mC polymerization charges. The corresponding ion-selective membranes were then fabricated and blade-coated onto the PEDOT: PSS-plated Au@Cu working electrode area. Here, the $Na^+$ selective membrane cocktail was prepared by mixing Na ionophoreX (1 mg), sodium tetrakis [3,5-bis(trifluoromethyl)phenyl] borate (NaTFPB) (0.55 mg), polyvinyl chloride (PVC) (33 mg), bis(2-ethylehexyl) sebacate (DOS) (65.45 mg) in 660 uL tetrahydrofuran, while the $K^+$ selective membrane cocktail was prepared by mixing valinomycin (2 mg), Sodium tetraphenylboron (NaTPB) (0.5 mg), PVC (32.75 mg), and DOS (64.75 mg) in 350 μL cyclohexanone. 4 μL of as-prepared $Na^+$ and $K^+$-selective membrane cocktail was blade-coated onto the PEDOT: PSS-plated Au@Cu working electrode area. For the working electrode of the pH sensor, PANI was electrodeposited onto the Au@Cu working electrode area by a periodic voltage wave (an amplitude of 0.9 V, frequency of 1 Hz and duty cycle of 10% for 1200 cycles) in a fresh electrolyte containing 0.1 M aniline and 1 M HCl. Glucose and lactate sensors are also constructed based on two-electrode configuration, where Au@Cu electrodes were directly employed as the counter electrodes and the PB-modified layer as working electrodes. A PB mediator layer was deposited onto the Au electrodes by cyclic voltammetry from 0 V to 0.5 V (versus Ag/AgCl) for two cycles at a scan rate of 20 mV/s in a fresh solution containing 2.5 mM $FeCl_3$, 100 mM KCl, 2.5 mM $K_3Fe(CN)_6$, and 100 mM HCl for both amperometric sensors. 1% chitosan solution was first prepared by dissolving chitosan in 2% acetic acid and magnetic stirring for about 1 h; next, the chitosan solution was mixed with single-walled carbon nanotubes (2 mg/mL) by ultrasonic agitation over 30 min to prepare a viscous solution of chitosan and carbon nanotubes. To prepare the glucose oxidase solution for the working electrode of the glucose sensor, the chitosan/carbon nanotube solution was mixed thoroughly with glucose oxidase solution (10 mg/mL in PBS of pH 7.2) in a volume ratio of 2:1. To prepare the lactate oxidase solution for working electrode of lactate sensor, the chitosan/carbon nanotube solution was mixed thoroughly with lactate oxidase solution (20 mg/mL in PBS of pH 7.2) in a volume ratio of 1:1. The glucose and lactate sensors were obtained by respectively blade-coated 3 μL of the glucose and lactate oxidase solution onto the Prussian blue-deposited Au@Cu electrode on both sides of the electrode area. The sensor arrays were then dried overnight at 4 °C before use.

## Assembly of the integrated in-textile sensing system

The integrated sensing system was fabricated by sewing the biosensor array and the in-textile circuit together to form a headband-shaped sensing patch. The circuit system adopted an integrated chip ESP32 as a microcontroller for multi-channel signal transmission and wireless display. The microcontroller for signal processing was programmed by the software Vscode. The potentiometric signal processing system (pH, $K^+$, and $Na^+$) and the amperometric signal processing system (glucose and lactate) mainly employed the LMV358 chip as a low-power operational amplifier with a rail-to-rail output swing. For the potentiometric signal processing, the extracted open circuit potential from the sensor was first amplified by the gain factor of 2–4 before

noise suppression. For the amperometric signal processing, the extracted direct current from the sensor was first amplified by the gain factor of $10^6–10^7$ before noise suppression by the passive 2nd low-pass filter. Then the microcontroller processed and transmitted the data to the phone via the Bluetooth chip. A lithium-ion battery of 200 mAh can directly power the whole system at a nominal voltage of 3.7 V.

## On-body sweat analysis

For real-time biomarker monitoring, the headband was connected to the smartphone App via Bluetooth. Volunteers were first asked to wear the headband and cycled for around 10 min as a warm-up process. After the sweat fills the cavity, the volunteers perform a 20-min static biking. Meanwhile, the App connection started recording data, and decoded sweat analysis results were displayed on the cellphone. Sweat was simultaneously collected in 800 s, 1200 s, and 1600 s to compare sensor data with measurements from pH meter, ICP-MS, and HPLC.

## Ex-situ evaluation of the sweat samples

Ex-situ sweat sensing was also conducted with exercise-induced sweat collected from the subjects' foreheads near the epidermal areas where the sensing headband was worn. Sweat samples were collected with microtubes every 400 s. The areas were wiped and cleaned with gauze after sample collection. The sweat samples were temporarily stored in the refrigerator at −4 °C for further ICP-MS and HPLC validation.

## Characterization

The optical microscopy images and movies were captured by NIKON Eclipse Ni-U. Morphologies of samples were also observed by the scanning electron microscope (SEM, TESCAN MAIA3, Brno, Czech Republic). The mechanical properties of the conductive metal patterns in fabrics were evaluated by using an Instron 5566 universal testing system. The electrical properties of the samples under different bending states were investigated by a Keithley 2400 Sourcemeter coupled with a computer-controlled stretching motor. The air permeability value (mm/s) of all the samples was tested according to ASTM D737-08 using a MO21S air permeability tester (SDL Americ, Inc.). Moisture permeability tests for the samples were performed using the cup method according to the textile standard E96/E96M-13. The moisture transmission rate $(g/m^2/day)$ was determined by measuring the weight loss of the water vapor in a cup with its opening firmly covered by the tested specimen. Both air permeability and moisture permeability tests were performed at around 26 °C and 65% relative humidity (testing duration of 2 weeks). The washing tests for the samples followed the AATCC 135 test standard, where samples were washed with 1.8 kg loading in a Whirlpool WTW4955H washing machine. Electrodeposition and sensor performance were performed by the electrochemical workstations (CHI660e). The electrochemical characterization of the sensor was evaluated in PBS-based solutions (pH 7.0) by changing the concentrations of analysts, except for the pH sensor. As for pH, $Na^+$, and $K^+$ sensors, direct recording of OCP from the two-electrode system was adopted. Chronoamperometry was utilized to characterize the lactate and glucose working electrodes during optimization (vs. Ag/AgCl). The pH sensing values were validated by the commercial pH meter (PHS-3C). The $Na^+$ and $K^+$ concentrations in the collected sweat samples were validated by using ICP-MS (Thermo Q Exactive), while the glucose and lactate concentrations were via HPLC (Shimadzu LC-2010C).

## Human participant recruitment

All experiments were performed according to the university guidelines (The Ethics Guidelines for Research Involving Human Subjects or Human Tissue from Southern University of Science and Technology, SUSTech Institutional Review Board, 20200037). The evaluation of the fully integrated biosensing headband in sweat samples for human participants followed all the ethical regulations according to the

protocol above. The participants (age range 20–30 years) were recruited from SUSTech campus through advertisement by posted notices. All participants gave written informed consent before participation in the study. The authors affirm that human research participants provided informed consent for publication of the images in Fig. 5 and the movie in Supplementary Movie 2.

## Reporting summary
Further information on research design is available in the Nature Portfolio Reporting Summary linked to this article.

## Data availability
The data that support the findings of this study are available in this article and supplementary materials. All raw and analyzed datasets generated during the study are available from the corresponding authors on request.

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

## Acknowledgements

P.W.W. and X.H.M. contributed equally to this work. The authors acknowledge the financial support from the State Key Laboratory of Ultra-precision Machining Technology (Grant No. 1-BBXR to Z.J.Z.), the RGC Senior Research Fellow Scheme (Grant No. SRFS2122-5S04 to Z.J.Z.), and the Hong Kong Polytechnic University (1-CD44 to Z.J.Z., 1-ZVT8 to Q.Y.H) This work was also supported by the National Natural Science Foundation of China (62201243 to Y.J.L., 52203318 to Q.Y.H), the Fundamental and Applied Research Grant of Guangdong Province (2021A1515110627 to Y.J.L.), Shenzhen Stable Support Plan Program for Higher Education Institutions Research Program (NO. 20220815153728002 to Y.J.L.).

## Author contributions

Conceptualization: P.W.W., Q.Y.H., Y.J.L., Z.J.Z. Idea discussion: P.W.W., X.H.M., Z.Q.L., H.L.X., Z. J. C., H.H., Q.Y.H., Y.J.L., Z.J.Z. Experiment: P.W.W., X.H.M., F.C., H.L.X., Q.Y.H., Z.Q.L., X.Y.Z., Y.Q.S., H.H. Supervision: Q.Y.H., Y.J.L., Z.J.Z. Data analysis: P.W.W., X.H.M., Q.Y.H., Y.J.L., Z.J.Z. Writing-original draft: P.W.W., X.H.M., Z.J.Z., Y.J.L., Q.Y.H. Writing-review and editing: P.W.W., X.H.M., Z.J.Z., Y.J.L., Q.Y.H.

## Competing interests

The authors declare no competing interests.
