## [Peer Review File · Nature Communications]

Reviewers' comments:

Reviewer #1 (Remarks to the Author):

This manuscript outlines the implementation of micrometer-scale metallic electrodes in fabrics using an in-textile photolithography process. In contrast to traditional fiber electronics, in-textile photolithography simplifies the process of metal patterning, eliminating the need for complex weaving or connecting conductive fibers to integrate the desired electronic circuits into textiles. The in-textile photolithography process is intriguing and holds promise for the development of next-generation wearable electronics. However, there are concerns regarding the applicability of this patterning process for universal use in electronic applications on or within fibers.

The in-textile photolithography process depicted in Figure 1a appears to require an etching step to pattern the metal electrodes. While wet etching is a conventional and widely used semiconductor process, it may not be the most suitable choice for achieving precise patterning. From a microelectronic perspective, it's challenging to consider a 100 μ m scale as high resolution. In this regard, the use of a process like PAMD (Polymer-Assisted Metal Deposition) seems more appropriate for forming in-textile metallization. If the author were to propose a selective patterning PAMD process without the need for etching, it could provide a better opportunity for implementing wearable electronics with improved precision.

The authors primarily emphasize the direct patterning of metal electrodes through in-textile photolithography. In previous research on fiber electronics, other studies have demonstrated the integration of multiple electronic components, including transistors, inverters, ring oscillators, UV sensors, temperature sensors, and even biosensors, on monofilament fibers and textiles. Therefore, the authors should highlight the impact of the in-textile photolithography process by demonstrating the integration of these diverse electronic components, rather than simply showcasing a series of connections to conventional rigid chips on textiles.

Reviewer #2 (Remarks to the Author):

This manuscript reported a photolithography approach to fabricate precise metal patterns on textile substrates with desirable conductivity, stability, and permeability. The controllable coating approach was beneficial for integrating multifunctional electronics into textiles. However, the process of the in-textile photolithography approach seems relatively complex and time-consuming, and in which many organic solvents/chemicals were used, likely leading to degradation of wearable safety (and even toxic) and

physical properties of textiles. Also, the approach discussed in this manuscript focused on the polyester-based textile, I doubt if it is suitable for textiles made of other typical fibers, like cotton, nylon, vinylon and spandex. Some other comments are listed below for reference.

1. The SEM images in Fig. 1c and Supplementary Fig. 2 presented that there were some Cu particles on the polyester fabric after polymer-assisted metal deposition (PAMD), indicating uneven Cu deposition. I am not sure whether this non uniformity adversely affects the conductivity of the patterned area?

2. Fig. 2d demonstrated that the metal pattern exhibited stability to withstand repeated bending cycles. However, deformations such as stretching, twisting, and bending with different curvature radii, are also likely to occur in practical wearable applications. More data about stability and durability will be helpful to better prove the validity of the method.

3. The textiles with different cover factors should influence the uniformity and integrity of deposited metal films on different fabrics. Please provide more detailed comparisons and discussion.

4. The surface morphology of metal deposited on textiles showed significant impact on the performances of the device. Please provide more data to explain this result.

5. Whether the metal pattern deposited on textile cause degradation of conductivity compared with the counterparts on thin film substrates?

6. As for double-sided UV exposure procedure, what is the UV exposure depth? what is the thickness of the textile? how about the properties for the thicker textiles?

Reviewer #3 (Remarks to the Author):

The manuscript presents a well-written study on high-resolution in-textile photolithography patterns. The authors claim that the key innovation involves integrating the PAMD process with two-sided lithography to enhance precision in in-textile lithography patterns. This study elucidates its underlying principles, provides ample experimental evidence, and aptly demonstrates its applications. However, there is room to justify the novelty of this study. Here are some comments to be addressed.

1. The key technology presented in this work is the combination of two common technologies: PAMD and photolithography. PAMD for textile deposition is a well-established, mature process, as evidenced by numerous references, such as Li P. et al. Polymer-assisted metal deposition (PAMD) for flexible and wearable electronics: principle, materials, printing, and devices. *Advanced Materials* 31.37 (2019): 1902987. Moreover, double-sided photolithography has been previously reported in many research articles, such as Pal P. et al. Front-to-back alignment techniques in microelectronics/MEMS fabrication: a review. *Sensor Letters* 4.1 (2006): 1-10. The authors are suggested to strongly justify the novelty of the "PAMD + photolithography" process.

2. "Line 135 ... with the patterning resolution ranging from 100 μm to 3 mm, demonstrating the wide applicability of in-textile photolithography as a universal method in creating high-resolution, multifunctional, and highly integrated circuits for textile electronic applications." This paragraph needs further clarification. Compared to the capabilities of photolithography, a 100-micron resolution is acceptable, even though not very high, whereas a 3 mm resolution cannot be classified as "high resolution". For various metal materials (i.e., Cu, Ag, Ni), maintaining a consistent resolution of approximately 100 microns should be considered a basic but appealing objective.

3. The demonstration of the integrated in-textile sensing headband is interesting. Where is the battery? Also, more characterizations about the resulting headband should be provided, such as the long-term stability, washability, etc.

Point-to-point response to reviewers' comments

Reviewer #1:

This manuscript outlines the implementation of micrometer-scale metallic electrodes in fabrics using an in-textile photolithography process. In contrast to traditional fiber electronics, in-textile photolithography simplifies the process of metal patterning, eliminating the need for complex weaving or connecting conductive fibers to integrate the desired electronic circuits into textiles. The in-textile photolithography process is intriguing and holds promise for the development of next-generation wearable electronics. However, there are concerns regarding the applicability of this patterning process for universal use in electronic applications on or within fibers.

Response: We thank the reviewer for carefully reading our manuscript and the encouraging feedback. The comments have been carefully addressed as follows.

1. The in-textile photolithography process depicted in Fig. 1a appears to require an etching step to pattern the metal electrodes. While wet etching is a conventional and widely used semiconductor process, it may not be the most suitable choice for achieving precise patterning. From a microelectronic perspective, it's challenging to consider a 100 μm scale as high resolution. In this regard, the use of a process like PAMD (Polymer-Assisted Metal Deposition) seems more appropriate for forming in-textile metallization. If the author were to propose a selective patterning PAMD process without the need for etching, it could provide a better opportunity for implementing wearable electronics with improved precision.

Response: We thank the reviewer for raising this comment. Patterning high-resolution metal electrodes in textiles has been technically challenging in the past two decades in the development of e-textiles (Nature Communications, 2017, 8: 1202; Advanced Engineering Materials, 2023, 25: 2201548). Directly printing typically results in low resolutions, in the range of 400-500 \$\mu\text{m}\$ (Nature Communications, 2022, 13: 2190; Chemical Engineering Journal, 2022, 429: 132196). To increase the patterning resolution, some researchers have developed methods to first coat the rough textile surfaces with additional epoxy-like planarization layer, followed by patterning of metal electrodes. Although this approach can effectively improve the resolution, the planarization process turned the porous textile structure into a dense thin-film like substrate, and then lose the permeability and wearing comfort of textiles. In this work, we developed in-textile photolithography to address the challenge. We demonstrated that 100 \$\mu\text{m}\$ resolution is reproducibly achieved with our approach, and the as-made electrodes are highly conductive and flexible. The E-textiles maintained their original porous structure, permeability, and textile-like comfort.

We would like to emphasize that the resolution of in-textile photolithography is largely limited by the fabric structure. In the manuscript, we have demonstrated the ability to achieve 100 \$\mu\text{m}\$ on a fabric made of 100 \$\mu\text{m}\$ yarns, which are considered to be one of the finest yarns in textiles. To the

best of our knowledge, this resolution, without sacrificing permeability, is the highest among the literature reports.

We have previously reported selective patterning PAMD processes (Advanced Materials, 2013, 25: 3343-3350; Advanced Materials, 2016, 28: 4926–4934) that yielded much worse resolutions (~500 μm) on textiles. These direct patterning processes had serious diffusion issues, largely associated with the capillary effect of textile fibers and the isotropic growth of metal during the PAMD process. In this work, we intentionally used PAMD to grow a high-quality coating first, followed by in-textile photolithography and wet selective etching process to avoid the abovementioned problems, so that the metal electrodes we developed in the work showed much higher resolution.

We revised the relative discussion in the introduction section:

“Nevertheless, due to the rough fabric surfaces and ink diffusion along the fibers within the 3D textile structures, such an on-textile patterning method is challenging to achieve a highly conductive and robust conductive patterns with precise linewidth thinner than 0.5 mm. Moreover, the conductive pastes largely covering the fabric surfaces will not only block the air and moisture permeability of textiles but also stiffen the fabrics by binding the soft fiber bundles, which may lead to cracks or even delamination with mechanical interferences⁴¹⁻⁴⁴. To increase the patterning resolution, some researchers have developed methods to first coat the rough textile surfaces with additional epoxy-like planarization layer, followed by patterning of metal electrodes. Although this approach can effectively improve the resolution, the planarization process turned the porous textile structure into a dense thin-film like substrate, and then lose the permeability and wearing comfort of textiles. In short, current state-of-the-art patterning methods for textiles show limitations in simultaneously offering conductive patterns with high resolution, excellent electrical conductivity, and mechanical robustness with retained permeability and softness.

To address these challenges, we herein propose an in-textile patterning technology by combining the PAMD and double-sided photolithography, named in-textile photolithography, to create unique 3D interconnected high-resolution and robust metal patterns in textiles. The in-textile photolithography allows for high-resolution and highly precise deposition of metal patterns on fabrics without the need for a binder and maintains the 3D porous structure of the textiles. Thus, the as-prepared E-textiles could maintain their desirable air and moisture permeability, flexibility and wearing comfort. The exceptionally high resolution of 100 μm for metal patterning on the fabric made of 100 μm yarns was achieved, which overcomes the key limitations of textile pore sizes and the diffusion issue due to capillary effect in the yarn structures. Attributed to the high-resolution property of this in-textile patterning method, fine interconnects and electrode patterns in textiles can be realized, fulfilling the requirements of commercial chip integration and contributing to high-performance and miniaturized device development. Notably, the metal patterns permeate inside the textile scaffold and exhibit electrical conduction throughout the fabric

thickness within the patterned area, enabling the construction of electronics on both sides of the fabric and showing promise for multi-layer circuit construction on a single piece of textile. The as-fabricated in-textile metal patterns also deliver outstanding bending stability of 10,000 times, and washing stability of 20 times with negligible conductivity variation.” (Page 3)

2. The authors primarily emphasize the direct patterning of metal electrodes through in-textile photolithography. In previous research on fiber electronics, other studies have demonstrated the integration of multiple electronic components, including transistors, inverters, ring oscillators, UV sensors, temperature sensors, and even biosensors, on monofilament fibers and textiles. Therefore, the authors should highlight the impact of the in-textile photolithography process by demonstrating the integration of these diverse electronic components, rather than simply showcasing a series of connections to conventional rigid chips on textiles.

Response: We thank the reviewer for pointing out this issue. we highlighted the impact of the in-textile photolithography process by demonstrating the construction of multifunctional electronic devices, including biosensors (pH, K^+ , Na^+ , glucose and lactate) (Fig. 4) and energy storage devices (supercapacitors) (Fig. 3d). Besides, we also emphasize that the highly controllable and conformal metal coating on fibers provides electrical conductivity throughout the fabric thickness. Thus, multi-layered electronic devices integration on a single piece can also be realized (Fig. 3e). Furthermore, the integrated E-textile for wireless sweat monitoring consists of multiple electronic devices including transistors, inverters, and electrochemical sensors for wearable applications (Fig.5a and b).

Fig. 3 High-performance electronics enabled by robust and high-resolution conductive metal patterns. **a** Resistance of the Cu patterns with different linewidths in polyester fabric during the 10,000-cycle bending test (bending radius: 4.4 mm). **b** Resistance changes of the Cu patterns with and without additional Au deposition upon 20 washing cycles. **c** Cyclic voltammety curves of the micro-supercapacitors made with interdigital Ni electrodes with different linewidths. MnO₂ is the electrochemically active material. **d** Areal capacitance of the micro-supercapacitors with different linewidths. Insets are digital images showing the electrode arrays of the micro-supercapacitors with different linewidths. **e** Digital images of the double-sided wearable temperature monitoring patch with *in-situ* alarming function based on high-resolution and double-sided Cu pattern in polyester fabric.

Fig. 4 Fabrication and performance evaluation of the multiplexed biosensor array for sweat monitoring. **a** Schematic illustration showing the fabrication process of the biosensor array in fabric. **b** Digital image of as-prepared biosensor array in fabric. **c-g** Sensing performance of as-made pH, Na⁺, K⁺, glucose, and lactate sensors.

Fig. 5. Design and on-body application of the integrated in-textile headband. **a** Digital image of the in-textile headband integrated with multiplexed sensing arrays and data analysis transmission circuit. **b** The logic flow of the in-textile headband. **c** Digital image showing *in-situ* sweat monitoring of the in-textile headband during exercise (the headband is mounted on the forehead). **d** Custom-designed mobile application for real-time and continuous biomarker sensing. **e** Representative real-time *in-situ* sweat biomarkers concentration monitoring during endurance cycling with *ex-situ* validation via analytical tools (pH meter, ICP-MS, and HPLC).

Reviewer #2 (Remarks to the Author):

This manuscript reported a photolithography approach to fabricate precise metal patterns on textile substrates with desirable conductivity, stability, and permeability. The controllable coating approach was beneficial for integrating multifunctional electronics into textiles. However, the process of the in-textile photolithography approach seems relatively complex and time-consuming, and in which many organic solvents/chemicals were used, likely leading to degradation of wearable safety (and even toxic) and physical properties of textiles. Also, the approach discussed in this manuscript focused on the polyester-based textile, I doubt if it is suitable for textiles made of other typical fibers, like cotton, nylon, vinylon and spandex. Some other comments are listed as below for reference.

Response: We thank the reviewer for the kind suggestions. Regarding the concerns on biocompatibility, we further conducted the long-term on-skin test to evaluate the wearing safety. The as-fabricated sensor electrodes did not show observable effects on the skin while other traditional flexible substrates, such as polyester film, led to obvious skin erythema at the covered skin caused by impermeable structure and high adhesion strength. We include discussions on the biocompatibility below:

“To evaluate the wearing safety, we conducted a long-term on-skin test. As shown in Supplementary Fig. 19, the as-fabricated sensor electrodes did not show observable effects on the skin while other traditional flexible substrates, such as polyester film, led to skin erythema possibly due to the impermeable structure and large adhesion strength.” (Page 16)

Supplementary Fig. 19. Biocompatibility of the sensor electrodes showing the skin irritation effects on the forearm which was covered with the in-textile sensor electrodes, polyester (PET) film, PDMS film, and commercial medical tape.

We also evaluate whether the organic solvents/chemicals would affect the physical properties of textiles. We include discussions in the manuscript as follows:

“As shown in Supplementary Fig. 7, the tensile strain at break of the polyester fabric after in-textile photolithography process has decreased about 11 % compared to the original commercial polyester fabric, which indicates that the physical properties of the textiles can be well-maintained.” (Page 8 -9)

Supplementary Fig. 7. Tensile properties of the original commercial polyester fabric and the one after in-textile photolithography.

In addition, we have carried out the in-textile lithography process on other textile substrates to clarify the versatility concern of the reviewer. The as-developed approach can be applied to a variety of textiles, including polyester-based textiles, glass fiber and cotton (Fig. 1h and Supplementary Fig. 3). Among these, it is found that the polyester-based textiles have even texture at the macro level and smooth fiber surface at the micro level, enabling the controllable electroless deposition of active materials on the fabric surface.

Fig. 1 In-textile photolithography for conductive metal patterns in fabrics. **a** Schematic illustration showing the development of textile electronics by in-textile photolithography technology. **b** Digital image of the Cu PCB circuit patterned in polyester fabric. Inset is the high-resolution image showing the Cu tracks in the fabric. **c** SEM image showing the boundary of the Cu pattern in polyester fabric. **d** Cross-sectional SEM image showing the boundary of the Cu pattern in polyester fabric. **e** Digital image showing the water vapor permeability and softness of the fabric patterned with Cu patterns. **f** Digital image of the Ag PCB circuit patterned in polyester fabric. **g** Digital image of the Ni interdigital electrodes with different scales patterned in polyester fabric. **h** Digital image of the Cu sensing electrode arrays patterned in glass-fiber fabric. **i** Digital image of the Ni electrode patterns in non-woven PP fabric.

Supplementary Fig. 3. Digital image of the Cu electrode patterns in cotton fabric.

1. The SEM images in Fig. 1c and Supplementary Fig. 2 presented that there were some Cu particles on the polyester fabric after polymer-assisted metal deposition (PAMD), indicating uneven Cu deposition. I am not sure whether this non uniformity adversely affects the conductivity of the patterned area?

Response: We thank the reviewer for the question. Normally, nanoparticles on thin films prepared via sputtering and vacuum evaporation can be observed under high-resolution characterization (Surface & Coatings Technology, 2016, 291: 286-291). The deposited copper surface shows similar typical nanoparticle structures, which is macroscopical uniform and has a negligible impact on conductivity.

2. Fig. 2d demonstrated that the metal pattern exhibited stability to withstand repeated bending cycles. However, deformations such as stretching, twisting, and bending with different curvature radii, are also likely to occur in practical wearable applications. More data about stability and durability will be helpful to better prove the validity of the method.

Response: We thank the reviewer for the suggestions. More data about stability and durability was collected to better prove the validity of the washability and long-term stability when crumpling the circuit interconnect, as shown in Supplementary Fig. 18.

“To further validate the robustness of the in-textile integrated circuits, washability and mechanical stability were characterized. With Ecoflex encapsulation, the interconnect resistance variation measured by connecting to 0 Ω resistors can be reduced to 50 % after 20 cycles of washing (Supplementary Fig. 18a). It is also observed that the resistance of the interconnects did not show

obvious change (11% variation) within the 120 cycles of crumpling test (Supplementary Fig. 18b).” (Page 16)

Supplementary Fig. 18. Electrical performance of circuit interconnects. (a) Resistance upon 20 washing cycles (encapsulated with Ecoflex and measured by connecting the interconnects to 0 Ω resistors). (b) Resistance of the interconnects during 180 times crumpling test.

3. The textiles with different cover factors should influence the uniformity and integrity of deposited metal films on different fabrics. Please provide more detailed comparisons and discussion.

Response: We thank the reviewer for pointing out this issue. The uniformity and integrity of deposited metal film are determined by the solution concentration and the reaction time in the PAMD process, which is independent of textiles with different cover factors. The textiles with different cover factors do not influence the uniformity and integrity of deposited metal films on different fabrics (Advanced Materials, 2019, 31: 1902987).

4. The surface morphology of metal deposited on textiles showed significant impact on the performances of the device. Please provide more data to explain this result.

Response: We thank the reviewer for the comment. Indeed, the surface morphology of metal deposited on textiles showed significant impact on the device performances. Our method preserves the unique three-dimensional porous structures of the fabric to achieve uniform metal coating. Based on theoretical calculation, with such conformal metal coating on the textile fiber, the surface area is 370 % larger than the one via screen printing (the planarization process). We prepared micro-supercapacitors to demonstration the superior of such methods for device performance enhancement. As shown in Fig. 3d, it contributes to an areal capacitance enhancement of more

than 300 % for micro-supercapacitors with the same electroplating conditions, showing a significant impact on the micro-supercapacitors performance. We included the discussion in the results and method sections.

“Compared to the on-textile patterning approach (such as screen printing) with limited resolution of 400 μm and resulted thin film materials coverage on one side of the electrodes on the fabric, the in-textile photolithography technique could create interdigital electrodes that preserve the unique three-dimensional porous structures. Uniform and high-resolution metal coating with both linewidth and their interspace down to 200 μm could ensure successful device fabrication without short circuit issue among electrodes. Based on theoretical calculation, with such conformal metal coating on the textiles, the surface area is estimated to be 370 % larger than the thin films formed via screen printing. As a result, under the same footprint area, the in-textile interdigital electrodes could offer a larger surface area for the conformal deposition of electrochemically active materials (i.e., manganese dioxide (MnO_2) in this work). The as-fabricated manganese dioxide (MnO_2) micro-supercapacitors deliver significantly areal capacitance enhancement of more than 300 % compared to those made by screen printing under the same electroplating conditions (Fig. 3c, d and Supplementary Fig. 9).” (page 12)

Fig. 3 High-performance electronics enabled by robust and high-resolution conductive metal patterns. **a** Resistance of the Cu patterns with different linewidths in polyester fabric during the 10,000-cycle bending test (bending radius: 4.4 mm). **b** Resistance changes of the Cu patterns with and without additional Au deposition upon 20 washing cycles. **c** Cyclic voltammetry curves of the micro-supercapacitors made with interdigital Ni electrodes with different linewidths. MnO_2 is the electrochemically active material. **d** Areal capacitance of the micro-supercapacitors with different linewidths. Insets are digital images showing the electrode arrays of the micro-supercapacitors

with different linewidths. **e** Digital images of the double-sided wearable temperature monitoring patch with *in-situ* alarming function based on high-resolution and double-sided Cu pattern in polyester fabric.

“Surface area ratio simulation and calculation of the micro-supercapacitors

One elliptical conductive weft yarn (radius of ellipse: 2.5/1) has 12 semicircular conductive fibers while one elliptical conductive warp yarn (radius of ellipse: 3/1) has 12 semicircular conductive fibers patterned in *Polyester*_{0.98} fabric. The ratio calculation equation can be written as follows:

$$Ratio = \left(\frac{L_{weft} + L_{warp} \times \pi}{L} \right) \times 0.98 = \left(\frac{(n+2 \times 2) \times 2 \times 3 + (n+2 \times 3) \pi}{2 \times 4} \right) \times 0.98 = 3.72 \quad (1)$$

where *L* is the length of the cross section for one chosen yarn, *L_{warp}* and *L_{weft}* are the perimeters of chosen yarn, 0.98 is the cover factor of the fabric.” (Page 20)

5. Whether the metal pattern deposited on textile cause degradation of conductivity compared with the counterparts on thin film substrates?

Response: We thank the reviewer for pointing out the issue. The as-fabricated in-textile metallic Cu cloth showed superior conductivity up to $2.0 \times 10^7 \text{ S m}^{-1}$, which is typically in the same order of magnitude as the bulk Cu metal ($6.41 \times 10^7 \text{ S m}^{-1}$). Such electrical conductivity for high-resolution metal patterns in textiles fulfills the requirements for electronic device fabrication and integration. The slightly lower conductivity for the in-textile patterning is considered a trade-off for the desirable permeability for wearable bioelectronics.

6. As for double-sided UV exposure procedure, what is the UV exposure depth? what is the thickness of the textile? how about the properties for the thicker textiles?

Response: We thank the reviewer for the comment. We added the illustration and characterization as follows.

“The thickness of our as-prepared textile is 100 μm and the UV exposure depth is approximately 80 μm, a thickness that ensures the integrality of the Cu pattern after double-sided photolithography. Thicker textiles like Polyester with a thickness of 170 μm created through the single-sided UV exposure exhibited an average thickness of 80 μm (Supplementary Fig. 6). Noted that the textile within a thickness of 160 μm is suitable during the double-sided photolithography including our as-prepared textile.” (Page 8)

Supplementary Fig. 6. Cross-sectional optical image of the thicker Polyester textile fabricated via single-sided exposure. (textile with a thickness of 170 μm and photoresist with a thickness of 80 μm)

Reviewer #3 (Remarks to the Author):

The manuscript presents a well-written study on high-resolution in-textile photolithography patterns. The authors claim that the key innovation involves integrating the PAMD process with two-sided lithography to enhance precision in in-textile lithography patterns. This study elucidates its underlying principles, provides ample experimental evidence, and aptly demonstrates its applications. However, there is room to justify the novelty of this study. Here are some comments to be addressed.

Response: We thank the reviewer for carefully reading our manuscript and the encouraging feedback. The comments have been carefully addressed as follows.

1. The key technology presented in this work is the combination of two common technologies: PAMD and photolithography. PAMD for textile deposition is a well-established, mature process, as evidenced by numerous references, such as Li P. et al. Polymer-assisted metal deposition (PAMD) for flexible and wearable electronics: principle, materials, printing, and devices. *Advanced Materials* 31.37 (2019): 1902987. Moreover, double-sided photolithography has been previously reported in many research articles, such as Pal P. et al. Front-to-back alignment techniques in microelectronics/MEMS fabrication: a review. *Sensor Letters* 4.1 (2006): 1-10. The authors are suggested to strongly justify the novelty of the "PAMD + photolithography" process.

Response: We thank the reviewer for raising this comment. Patterning high-resolution metal electrodes in textiles has been technically challenging in the past two decades in the development of E-textiles (*Nature Communications*, 2017, 8: 1202; *Advanced Engineering Materials*, 2023, 25: 2201548). We have previously reported selective patterning PAMD processes (*Advanced Materials*, 2013, 25: 3343-3350; *Advanced Materials*, 2016, 28: 4926–4934) that yielded resolutions of 500 μm on textiles, which is much lower than the one achieved in our proposed method in this work. These direct patterning processes had serious diffusion issues, largely associated with the capillary effect of textile fibers and the isotropic growth of metal during the PAMD process.

In this work, we intentionally used PAMD to grow a high-quality coating first, followed by in-textile photolithography and wet selective etching process to avoid the abovementioned problems, so that the metal electrodes we developed in the work reach the highest record of textile conductive patterning of 100 μm . In this method, PAMD retains the porous and breathable structure of the textile, providing good conductivity and stability of the conductive pattern, while double-sided lithography contributes to the high-resolution for the pattern.

It is worth mentioning that our method is not a simple superposition of these two mature technologies. We further include schematic illustration below to justify the difference of our method compared to traditional double sided photolithography. In traditional double-sided lithography, both sides of the wafer are exposed simultaneously to achieve different patterns on both sides. While using double-sided lithography in textile, ultraviolet light would pass through

the porous metal textile prepared by PAMD into the interior of the textile, achieving complete protection of the metal coating inside the pattern area when etched. Thus, electrical conduction through the 3D textile porous structure can be achieved, enabling the construction of electronics on both sides of the fabric and showing promise for multi-layer circuit construction on a single piece of textile. Besides, the 3D porous structure of the metal pattern retained by PAMD has a larger surface area compared to other metallization methods, contributing to conformal active materials electrodeposition and device performance enhancement. Most importantly, the as-prepared E-textiles also maintained their original porous structure, permeability, flexibility and textile-like comfort.

The key innovation integrating the polymer-assisted metal deposition (PAMD) process with double-sided lithography to achieve practical E-textiles was illustrated in the manuscript:

“Nevertheless, due to the rough fabric surfaces and ink diffusion along the fibers within the 3D textile structures, such an on-textile patterning method is challenging to achieve a highly conductive and robust conductive patterns with precise linewidth thinner than 0.5 mm. Moreover, the conductive pastes largely covering the fabric surfaces will not only block the air and moisture permeability of textiles but also stiffen the fabrics by binding the soft fiber bundles, which may lead to cracks or even delamination with mechanical interferences⁴¹⁻⁴⁴. To increase the patterning resolution, some researchers have developed methods to first coat the rough textile surfaces with additional epoxy-like planarization layer, followed by patterning of metal electrodes. Although this approach can effectively improve the resolution, the planarization process turned the porous textile structure into a dense thin-film like substrate, and then lose the permeability and wearing comfort of textiles. In short, current state-of-the-art patterning methods for textiles show limitations in simultaneously offering conductive patterns with high resolution, excellent electrical conductivity, and mechanical robustness with retained permeability and softness.

To address these challenges, we herein propose an in-textile patterning technology by combining the PAMD and double-sided photolithography, named in-textile photolithography, to create unique 3D interconnected high-resolution and robust metal patterns in textiles. The in-textile photolithography allows for high-resolution and highly precise deposition of metal patterns on fabrics without the need for a binder and maintains the 3D porous structure of the textiles. Thus, the as-prepared E-textiles could maintain their desirable air and moisture permeability, flexibility and wearing comfort. The exceptionally high resolution of 100 μm for metal patterning on the fabric made of 100 μm yarns was achieved, which overcomes the key limitations of textile pore sizes and the diffusion issue due to capillary effect in the yarn structures. Attributed to the high-resolution property of this in-textile patterning method, fine interconnects and electrode patterns in textiles can be realized, fulfilling the requirements of commercial chip integration and contributing to high-performance and miniaturized device development. Notably, the metal patterns permeate inside the textile scaffold and exhibit electrical conduction throughout the fabric thickness within the patterned area, enabling the construction of electronics on both sides of the fabric and showing promise for multi-layer circuit construction on a single piece of textile. The as-fabricated in-textile metal patterns also deliver outstanding bending stability of 10,000 times, and washing stability of 20 times with negligible conductivity variation.” (Page 3)

2. “Line 135 ... with the patterning resolution ranging from 100 μm to 3 mm, demonstrating the wide applicability of in-textile photolithography as a universal method in creating high-resolution, multifunctional, and highly integrated circuits for textile electronic applications.” This paragraph needs further clarification. Compared to the capabilities of photolithography, a 100-micron resolution is acceptable, even though not very high, whereas a 3 mm resolution cannot be classified as "high resolution". For various metal materials (i.e., Cu, Ag, Ni), maintaining a consistent resolution of approximately 100 microns should be considered a basic but appealing objective.

Response: We thank the reviewer for pointing out the confusing part. Our approach maximumly preserves the advantages of the textile including the unique three-dimensional porous structure. Compared with the flexible thin film, it retains high flexibility, reliable wearing comfort, and excellent air and moisture permeability. Nevertheless, the yarn structure and pore size enclosed by the weft and warp yarns pose the key limitation on the patterning resolution and conductivity. For various metal materials, maintaining a consistent resolution of 100 μm is a basic objective on a flexible thin film (Science, 2023, 380: 735–742). We demonstrated that 100 μm resolution is reproducibly achieved in textiles with our approach, and the as-made electrodes are highly conductive and flexible (Fig. 2k). To the best of knowledge, the resolution of 100 μm is the highest record among reports in E-textiles especially those with weaving structure, compared to other traditional methods such as screen printing or catalyst transferring methods that normally reach a limit of 400-500 μm (Nature Communications, 2022, 13: 2190; Chemical Engineering Journal, 2022, 429: 132196). The interconnects resolution of 100 μm can fulfill the requirements of commercial chip integration. We previously indicated the patterning resolution range of from 100

μm to 3 mm to emphasize the patterning versatility. To clarify this issue, we revised the expression below:

“The in-textile photolithography allows for high-resolution and highly precise deposition of metal patterns on fabrics without the need for a binder and maintains the 3D porous structure of the textiles. Thus, the as-prepared E-textiles could maintained their desirable air and moisture permeability, flexibility and wearing comfort. The exceptionally high resolution of 100 μm for metal patterning on the fabric made of 100 μm yarns was achieved, which overcomes the key limitations of textile pore sizes and the diffusion issue due to capillary effect in the yarn structures. Attributed to the high-resolution property of this in-textile patterning method, fine interconnects and electrode patterns in textiles can be realized, fulfilling the requirements of commercial chip integration and contributing to high-performance and miniaturized device development. Notably, the metal patterns permeate inside the textile scaffold and exhibit electrical conduction throughout the fabric thickness within the patterned area, enabling the construction of electronics on both sides of the fabric and showing promise for multi-layer circuit construction on a single piece of textile. The as-fabricated in-textile metal patterns also deliver outstanding bending stability of 10,000 times, and washing stability of 20 times with negligible conductivity variation.” (Page 3)

“The 100 μm patterning resolution of the Ag circuit interconnects and Ni interdigitated electrodes in the polyester fabrics was achieved, demonstrating the wide applicability of in-textile photolithography as a universal method in the creation of high-resolution, multifunctional, and highly integrated circuits for textile electronic applications.” (Page 5)

Fig. 2 Characterization of high-resolution and conductive metal patterns in textiles. **a** Optical image (left), cross-sectional schematic diagram (top right), and SEM image (bottom right) of the Cu pattern fabricated by the in-textile photolithography method. **b** Optical image (left), cross-sectional schematic diagram (top right), and SEM image (bottom right) of the Cu pattern fabricated by on-textile patterning method (screen printing). **c** Comparison of water vapor and air permeability of the fabrics coated with Cu made by the in-textile photolithography method and the on-textile patterning method. **d** Comparison of resistance changes (R/R_0) of the Cu patterns made by the in-textile photolithography method and on-textile patterning method. Inset is the digital image showing the flexibility of the Cu patterns made by two methods. **e** Linear resistance of the Cu patterns as a function of the patterning resolution. **f** Schematic diagram of the double-sided UV exposure (left) and optical images of the face side and back side of the Cu pattern made by in-textile photolithography (right). **g** Schematic diagram of the single-sided UV exposure (left) and optical images of the face side and back side of the Cu pattern made by in-textile photolithography

(left). **h** Linear resistance of the Cu patterns as a function of the patterning resolution. **i** SEM images of the Polyester_{0.90}, Polyester_{0.94}, and Polyester_{0.98} fabrics. 0.90, 0.94, and 0.98 are the fabric cover factors of the corresponding fabrics. **j** Linear resistance of the Cu patterns as a function of patterning resolution. **k** SEM images showing the high-resolution Cu pattern with 100 μm linewidth (left) and 80 μm interspace between Cu patterns (right) in Polyester_{0.98} fabrics.

3. The demonstration of the integrated in-textile sensing headband is interesting. Where is the battery? Also, more characterizations about the resulting headband should be provided, such as the long-term stability, washability, etc.

Response: We thank the reviewer for the suggestions. We included the picture that shows the integration of the battery into the headband (Supplementary Fig. 15). The long-term stability of the headband can be evaluated in two aspects: stability of circuit interconnects and the biosensor electrodes. Supplementary Fig. 16b shows the conductivity of interconnects during mechanical crumpling cycles, demonstrating their mechanical robustness. Regarding the washing stability, the resistance of the Cu interconnects patterned in the fabric increased by around 3 folds (Fig. 3b) after 20 cycles of washing, while with Au deposition (Fig. 3b) or Ecoflex as encapsulation (Supplementary Fig. 16a), resistance variation can be reduced to only 50 %. Thus, the durability of integrated circuits made by in-textile photolithography can be ensured. Besides, we adopted biosensors as one of the functional device demonstrations in this work. The conductive patterns serve as the current conductor in the sensors, while the long-term sensing performance is mainly affected by the enzyme activity and unavoidable pollution from the skin surface (Nature Biomedical Engineering, 2022, 6: 1225–1235).

Supplementary Fig. 15. Digital image of the in-textile headband integrated with a commercial Li-ion battery.

Fig. 3 High-performance electronics enabled by robust and high-resolution conductive metal patterns. **a** Resistance of the Cu patterns with different linewidths in polyester fabric during the 10,000-cycle bending test (bending radius: 4.4 mm). **b** Resistance changes of the Cu patterns with and without additional Au deposition upon 20 washing cycles. **c** Cyclic voltammetry curves of the micro-supercapacitors made with interdigital Ni electrodes with different linewidths. MnO₂ is the electrochemically active material. **d** Areal capacitance of the micro-supercapacitors with different linewidths. Insets are digital images showing the electrode arrays of the micro-supercapacitors with different linewidths. **e** Digital images of the double-sided wearable temperature monitoring patch with *in-situ* alarming function based on high-resolution and double-sided Cu pattern in polyester fabric.

“To further validate the robustness of the in-textile integrated circuits, washability and mechanical stability were characterized. With Ecoflex as encapsulation, the interconnect resistance variation measured by connecting to 0 Ω resistors can be reduced to 50 % after 20 cycles of washing (Supplementary Fig. 18a). It is also observed that the resistance of the interconnects did not show obvious change (11% variation) within the 120-cycle crumpling test (Supplementary Fig. 18b).” (Page 16)

Supplementary Fig. 18. Electrical performance of circuit interconnects. (a) Resistance upon 20 washing cycles (encapsulated with Ecoflex and measured by connecting the interconnects to 0 Ω resistors). (b) Resistance of the interconnects during 180 times crumpling test.

REVIEWERS' COMMENTS

Reviewer #1 (Remarks to the Author):

I carefully reviewed the response letter and thoroughly examined the revised supporting information. The authors have effectively addressed numerous questions posed by the reviewers. However, the revised manuscript lacks specific data inclusion. For instance, it appears that Supplementary Information Figure 18 could be integrated into either Figure 2 or Figure 3.

Furthermore, I still have reservations regarding the use of the term "High resolution" in the title. In Figure 2k, it is evident that the minimum size of the electrode on textile is 80 μ m. Is this genuinely considered high resolution? Perhaps, using "Well-defined" would be more appropriate, highlighting the merits of this work.

In conclusion, I strongly recommend that the authors incorporate more data into the main figures and reconsider the current title.

Then, It is publishable in Nature Communications.

Reviewer #2 (Remarks to the Author):

I appreciate the the authors' efforts to well address the reviewers's comments. In general, although the photolithography approach proposed in this study has some intrinsic limits, this study indeed provides a new thought to realizing precise conductive patterns in electronic textiles. Given this, I am glad to see its publication in Nature Communications.

Reviewer #3 (Remarks to the Author):

My comments have been well addressed. This submission is ready for publication.

Point-to-point response to reviewers' comments

Reviewer #1:

I carefully reviewed the response letter and thoroughly examined the revised supporting information. The authors have effectively addressed numerous questions posed by them reviewers. However, the revised manuscript lacks specific data inclusion. For instance, it appears that Supplementary Information Figure 18 could be integrated into either Figure 2 or Figure 3.

Furthermore, I still have reservations regarding the use of the term "High resolution" in the title. In Figure 2k, it is evident that the minimum size of the electrode on textile is $80\mu\text{m}$. Is this genuinely considered high resolution? Perhaps, using "Well-defined" would be more appropriate, highlighting the merits of this work. In conclusion, I strongly recommend that the authors incorporate more data into the main figures and reconsider the current title.

Response: We thank the reviewer for carefully reading our manuscript and appreciate the reviewer's encouraging feedback. The comments have been carefully addressed below.

First of all, we integrated SI Figure 7 and 18 into Fig 3 to incorporate more data into the main figures.

Fig. 3 High-performance electronics enabled by robust and well-defined conductive metal patterns. **a** Tensile property of the commercial polyester fabric and polyester fabric after in-textile photolithography. **b** Resistance of the Cu patterns with different linewidths in polyester fabric during the 10,000-cycle bending test (bending radius: 4.4 mm). **c** Resistance changes of the Cu patterns with and without additional Au deposition upon 20 washing cycles. Error bars represent the s.d. of the mean from five Cu patterns. **d** Resistance of the interconnects during 180 times crumpling test. **e** Resistance upon 20 washing cycles (encapsulated with Ecoflex and measured by connecting the interconnects to 0Ω resistors). **f** Cyclic voltammetry curves of the micro-supercapacitors made with interdigital Ni electrodes with different linewidths. MnO_2 is the electrochemically active material. **g** Areal capacitance of the micro-supercapacitors with different linewidths. Insets are digital images showing the electrode arrays of the micro-supercapacitors with different linewidths. **h** Digital images of the double-sided wearable temperature monitoring patch with *in-situ* alarming function based on well-defined and double-sided Cu pattern in polyester fabric.

Then we replaced the term “high resolution” to “well-defined” in the title and other parts of the manuscript to highlighting the merits of our work.

“Well-defined in-textile photolithography towards permeable textile electronics” (Page 1)

“To address these challenges, we herein propose an in-textile patterning technology by combining the PAMD and double-sided photolithography, named in-textile photolithography, to create unique 3D interconnected well-defined and robust metal patterns in textiles.” (Page 3)

“The proposed innovative approach for well-defined metal patterns with sub-100 μm resolution in textiles will pave the way towards permeable, high-performance, multifunctional, and highly integrated textile electronics for practical wearable applications such as non-invasive monitoring and human-machine interfaces.” (Page 4)

“The fabrication of well-defined conductive metal patterns in textiles via utilizing in-textile photolithography is illustrated in Fig. 1a.” (Page 4)

“The 100 μm patterning resolution of the Ag circuit interconnects and Ni interdigitated electrodes in the polyester fabrics was achieved, demonstrating the wide applicability of in-textile photolithography as a universal method in the creation of well-defined, multifunctional, and highly integrated circuits for textile electronic applications.” (Page 5)

“Permeable, robust, and well-defined conductive metal patterns in textiles.” (Page 6)

“Most importantly, in-textile photolithography inherited the high precision of the conventional photolithography technique, enabling the creation of precise conductive patterns with sharp and well-defined boundaries.” (Page 7)

“To study the crucial role of double-side UV exposure in attaining well-defined conductive metal patterns in the textile scaffold, we compared the coating morphologies and electrical resistances of the metal patterns that were respectively subjected to single-sided and double-sided UV exposures during the fabrication.” (Page 8)

“**Fig. 2** Characterization of well-defined and conductive metal patterns in textiles.” (page 10)

“High-performance electronics enabled by robust and well-defined conductive metal patterns in textiles.” (Page 11)

“Specifically, these well-defined conductive patterns could benefit the development of high-performance miniaturized electronic devices.” (Page 12)

“**Fig. 3** High-performance electronics enabled by robust and well-defined conductive metal patterns.” (Page 13)

“**h** Digital images of the double-sided wearable temperature monitoring patch with in-situ alarming function based on well-defined and double-sided Cu pattern in polyester fabric.” (Page 14)

“Owing to the excellent mechanical robustness and permeability of the well-defined conductive patterns in textiles, the headband realizes real-time sweat collection and simultaneous monitoring of multiple sweat biomarkers with desirable wearing comfort.” (Page 19)